**communications** engineering

# Bridging mathematical modeling and AI for 3D coordinate recognition of moving objects without external reference and attitude measurement

Junfan Yi[1,2,3,5], Ke-ke Shang [2,5] ✉ & Michael Small[1,4]

Early positioning concentrates on static natural geographic features, shifting focus to capturing dynamic objects with the emergence of geographic information systems and the growing demand for spatial data. However, previous methods typically rely on expensive devices or external calibration objects for attitude measurement. Here we propose a real-time hybrid framework with a dual-phase strategy that leverages the time-series nature of dynamic objects, combining AI detection with mathematical modeling to estimate relative attitudes via efficient singular value decomposition, thus enabling reference-free 3D coordinate recognition. In particular, we enhance the state-of-the-art You Only Look Once version 12 model by incorporating time-series analysis for rapid and precise 2D detection, which serves as input for 2D-to-3D conversion via our singular value decomposition-based solver. By leveraging data from only three off-the-shelf smartphone cameras, the system achieves accurate and reference-free 3D positioning of a flying UAV. Experimental results demonstrate high precision in terms of RMSE, MAE, and R-squared. Therefore, under sensor-resource constraints, this AI-mathematics fusion enables real-time 3D coordinate recognition without traditional attitude measurement.

Geodetic positioning emerged from ancient geometric problems and subsequently evolved into a core technology of geomatics, applicable to any location worldwide, unrestrained by geographical boundaries. This evolution grants the technology systematic, dynamic, and interdisciplinary features[1], naturally drawing on insights from physics[2,3], mathematics[4–6], with a growing emphasis on artificial intelligence[7–10]. Crucially, rooted in both geometric challenges and spatial exploration demand, geodetic positioning is transforming from traditional ground-based methods to advanced spatial measurements. This transition, primarily from static geodetic mapping to dynamic 3D positioning, is driven by applied mathematics and computational science, and enriches geospatial information, which is essential for global environmental protection[11], climate change mitigation[7,12], and city sustainability[3,13], and further understanding the complex Earth system[2,7,10,14].

Currently, 3D positioning systems can be classified into two categories, active and passive. Active techniques, such as Global Navigation Satellite System (GNSS)[15], Inertial Navigation System[16] and Ultra Wide Band[17], involve the target actively determining its own 3D position by carrying signal reception devices. These methods, however, are inapplicable to non-cooperative targets, such as unidentified uncrewed aerial vehicles (UAVs). In contrast, passive methods, such as Light Detection and Ranging[18], photogrammetry[19], and Synthetic Aperture Radar[20], are utilized for non-cooperative targets but typically rely on external calibration objects or expensive attitude measurement devices to determine poses. To avoid reliance on attitude measurement devices, vision-based methods such as Structure from Motion (SfM)[21–24] have been developed to estimate camera poses and reconstruct 3D geometry directly from image sequences. However, SfM is fundamentally designed for scenarios with a moving camera observing a static scene, and requires rich environmental features for cross-view matching. In real-world scenarios, conventional SfM becomes ineffective when stationary cameras attempt to track a moving object against a

¹Complex Systems Group, Department of Mathematics and Statistics, The University of Western Australia, Crawley, Perth, WA, Australia. ²Computational Communication Collaboratory, Nanjing University, Qixia, Nanjing, Jiangsu, China. ³The School of Geography and Ocean Science, Nanjing University, Qixia, Nanjing, Jiangsu, China. ⁴Mineral Resources, CSIRO, Kensington, Perth, WA, Australia. ⁵These authors contributed equally: Junfan Yi, Ke-ke Shang.
✉e-mail: kekeshang@nju.edu.cn; keke.shang.1989@gmail.com

featureless sky, as there are insufficient distinctive features for correspondence (see Supplementary Discussion). Instead, we observe that the trajectory of the spatial target, captured as a position-time series across multiple cameras, can serve as a natural correspondence across views, effectively replacing the role of static scene features. Matching these sequential observations, geometric transformations can compute the relative poses (attitudes and positions) of cameras. In this perspective, we propose an efficient and simplified optical measurement system that utilizes cameras to establish relative spatial relationships through captured 2D coordinate time series (Fig. 1). By integrating rapid AI-driven 2D detection with a computationally efficient Singular Value Decomposition (SVD) solver, our framework requires only the known 3D positions of cameras, rather than their absolute attitudes, to achieve real-time, reference-free 3D positioning even with consumer-grade devices (Fig. 2). Specifically, we employ a dual-phase strategy (Algorithm 1): Phase I (Batch Initialization) accumulates observations to establish precise camera poses, while Phase II (Online Tracking) performs instantaneous coordinate transformation, jointly achieving high accuracy and real-time 3D coordinate recognition.

To verify the theoretical feasibility of the proposed framework, we conduct a numerical simulation in a $200 \times 200 \times 100$ m virtual 3D space (Fig. 3). Here, three ground-based cameras with unknown attitudes capture a flying object, represented as a 2D point in the images. During Batch Initialization, the captured 2D coordinate time series is used to estimate the relative poses between the three cameras using SVD. During Online Tracking, we combine the estimated poses with the 3D coordinates of the cameras to solve for the similarity transformation, which is subsequently used for real-time 3D coordinate recognition of the flying object. The resulting errors are negligible, primarily due to inherent numerical approximations in the simulation. Additionally, we consider engineering scenarios by introducing camera positioning errors, 2D coordinate deviations, and scene scaling factors to evaluate their impact on 3D coordinate recognition (Table 1; Fig. 4). We find that 2D coordinate deviations have a greater impact on the resulting errors compared to camera positioning errors. Moreover, a larger scene size amplifies the influence of both camera positioning errors and 2D coordinate deviations on 3D coordinate recognition. Nevertheless, the accuracy of 3D coordinate recognition remains within acceptable limits for general camera positioning and pixel deviation levels. The theoretical precision of our SVD-based coordinate transformation is further validated through simulation (see Supplementary Note 5), where errors are negligible under idealized conditions. Based on the 2D coordinate time series captured by multiple cameras, our mathematical approach for 3D coordinate recognition is precise, straightforward, and theoretically robust. This method efficiently converts 2D coordinates to 3D coordinates, making attitude measurement unnecessary.

Furthermore, to validate the proposed method in a real-world scenario, we conduct a UAV experiment within a $100 \times 100 \times 30$ m area, using a UAV as the flying object and employing three cameras for 3D coordinate recognition. In engineering applications, it is essential to establish a system that links object detection to real-world 3D coordinate recognition. Leveraging advancements in artificial intelligence[1], we efficiently detect the UAV as a bounding box and determine its centroid to derive 2D coordinates, which are then converted into 3D coordinates using our mathematical approach. Here, we employ YOLOv12 (You Only Look Once, version 12; Supplementary Note 3), a top-performing object detection method based on Convolutional Neural Networks (CNNs), to extract the detection bounding box of the UAV from the image. However, YOLOv12 still struggles with missed and false detections of small objects in complex scenarios[25]. To address these challenges, we propose YOLO-Time Series (YOLO-TS), which uses temporal sequences and a dual-phase design to effectively reduce missed and false detections. Using the detected 2D coordinate time series from the three cameras, we follow the steps outlined in simulation for 3D coordinate recognition. Additionally, to bridge theoretical modeling with engineering practice, we adopt Bundle Adjustment[26], a well-established numerical optimization technique in photogrammetry, to periodically refine camera poses and improve 3D positioning accuracy. Finally, the results of the 3D coordinate recognition show a Root Mean Square Error (RMSE) of 5.45 m, a Mean Absolute Error (MAE) of 4.83 m, and an R-squared value of 0.91 in a $100 \times 100 \times 30$ m scenario, demonstrating the robustness and effectiveness of our proposed time series-based method without prior knowledge of camera attitudes.

In summary, we establish a real-time hybrid framework with a dual-phase strategy for 3D coordinate recognition of non-cooperative targets that integrates AI-driven 2D detection and applied mathematics, requiring no attitude measurement. The core of our framework, which involves the transformation from 2D coordinates to 3D coordinates, leverages two key theories, time series and singular value decomposition, which ensure a robust and real-time transition from 2D to 3D.

## Results
### Numerical simulations of coordinate recognition
To validate the proposed real-time 3D coordinate recognition method, we conduct numerical simulations under the dual-phase strategy (Algorithm 1) using three non-aligned ground cameras and a moving point over 900 time

**Fig. 1 | The 2D-to-3D conversion framework.** The framework begins with AI-based 2D detection using a Convolutional Neural Network to obtain the 2D coordinate time series of the object in the captured images (see Supplementary Methods). These coordinates are then refined using time series and the physical feature of velocity. The core method utilizes time series and SVD to estimate the relative poses of the cameras. An SVD-based approach is further employed to calculate the similarity transformation matrix, deriving the camera-to-world coordinate transformation and ultimately achieving 3D geodetic positioning in the world coordinate system.

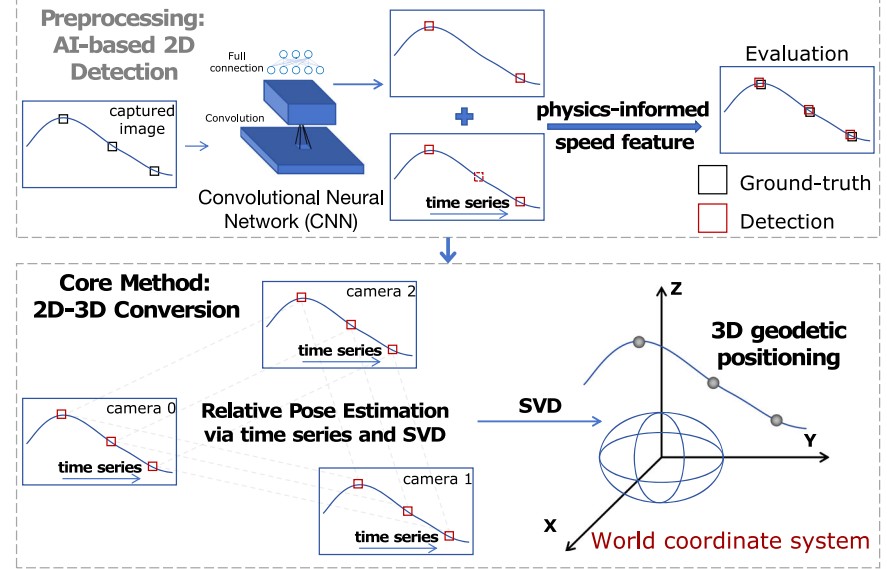

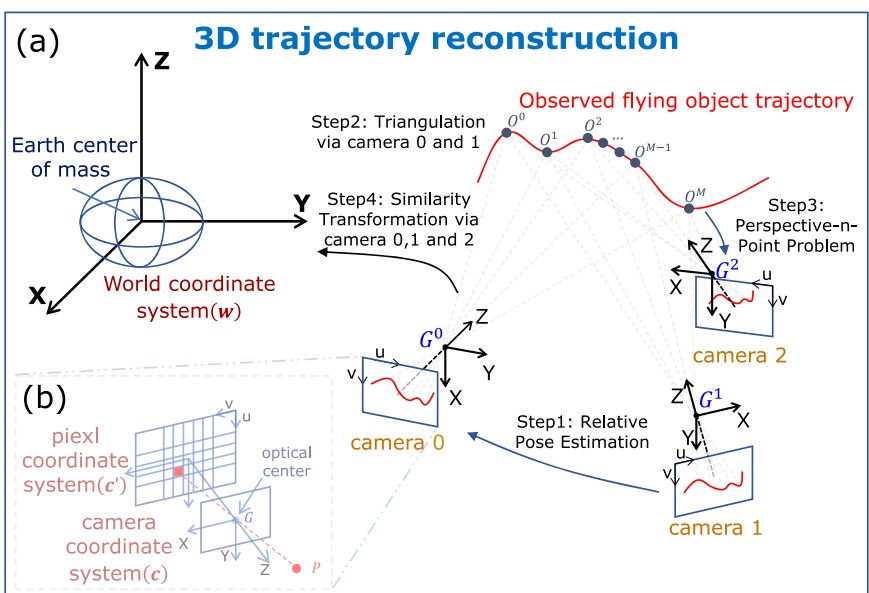

**Fig. 2 | Principle of 3D trajectory reconstruction.** In this paper, we utilize only three cameras to capture the trajectory of a flying object without measuring the attitudes of the cameras using calibration objects or devices. As shown in (**a**), step 1: Three ground-based cameras (Supplementary Methods) are directed towards the flying object, capturing its trajectory denoted as $O_{c'^i}^j$ at time step $j$ in the pixel coordinate systems of camera $i$ ($c'^i$). The projection of a point $P$ in 3D space onto the 2D pixel coordinate system is described as the pinhole camera model (**b**). Upon acquiring the 2D trajectories $O_{c'^0}^j$ and $O_{c'^1}^j$ in the pixel coordinate systems of camera 0 ($c'^0$) and camera 1 ($c'^1$), we employ a relative pose estimation method to calculate the relative pose of camera 1 with respect to camera 0. Step 2: With the 2D trajectories $O_{c'^0}^j$ and $O_{c'^1}^j$, and the determined pose of camera 1 relative to camera 0 from step 1, triangulation is used to calculate the 3D position of $O^j$ in the camera 0 coordinate system ($c^0$). Step 3: Having obtained the trajectory of $O^j$ in the camera 0 coordinate system ($c^0$), we solve the Perspective-n-Point problem to calculate the relative pose of camera 2 with respect to camera 0. Step 4: Given the coordinates of the three cameras $G^i$ in both the world coordinate system ($w$) and the camera 0 coordinate system ($c^0$), we solve for the similarity transformation between these two coordinate systems to compute the trajectory of $O^j$ in the world coordinate system ($w$).

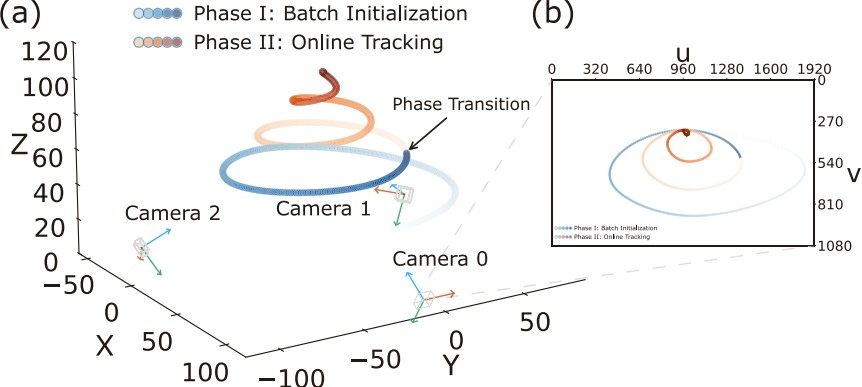

**Fig. 3 | Simulated scene for dual-phase validation. a** A spiral ascending trajectory is defined by 15 control points and interpolated via cubic splines to 900 frames at 30 FPS, spanning 70 m horizontally and ascending from 30 to 100 m altitude. The trajectory is divided into two phases: blue gradient represents Phase I (frames 0–299); orange gradient represents Phase II (frames 300–899). The arrow indicates the phase transition point. Three ground cameras, each located 100 m from the origin, are positioned at 120° intervals and oriented toward a focal point at (0, 0, 65) m. The optical centers in the world coordinate system are (100.0, 0.0, 0.0) m, ( − 50.0, − 86.6, 0.0) m, and ( − 50.0, 86.6, 0.0) m, respectively. The cameras are distortion-free, with focal lengths $f_x = f_y = 1000$ pixels, principal points at $c_x = 960$ pixels, $c_y = 540$ pixels, and pixel skew $s = 0$. **b** Projection of the 3D trajectory onto the image plane of Camera 0, showing the 2D pixel coordinates as observed by the camera. The axes $u$ and $v$ represent the horizontal and vertical pixel coordinates, respectively.

steps within a virtual 3D space measuring $200 \times 200 \times 100$ m (Fig. 3). The cameras are used to capture the 2D coordinates of the moving point at each time step. Additionally, the 3D coordinates of each camera in the world coordinate system are known.

The trajectory is divided into two phases. During Phase I (Batch Initialization), the first $N = 300$ frames are buffered. First, leveraging the time series of 2D coordinates from camera 0 and camera 1 over 300 time steps (with 8 time steps being the theoretical minimum, see Supplementary Methods), we obtain 300 pairs of 2D coordinates of the moving spatial point.

These pairs are used with SVD to extract the relative poses of camera 0 and camera 1 (Fig. 2a, step 1; Supplementary Methods). Second, with the calculated relative poses of these two cameras, we construct rays from the optical center to the 2D pixel points in both cameras. Applying SVD to minimize the distance between the target 3D point and the two rays, we obtain the 3D coordinates of the spatial point in the coordinate system of camera 0 (Fig. 2(a), step 2; Supplementary Methods). Third, for camera 2, based on the calculated 3D coordinates of the spatial point in the camera 0 coordinate system and its 2D coordinates in the pixel coordinate system of

**Table 1 | Sensitivity analysis of dual-phase simulation for 3D coordinate recognition**

| Error Source | Level | RMSE (m) | MAE (m) | Max (m) |
|---|---|---|---|---|
| Detection (pixel) | 1 | 1.19 ± 0.67 | 1.17 ± 0.67 | 1.86 ± 0.96 |
| | 2 | 1.78 ± 1.09 | 1.74 ± 1.08 | 2.86 ± 1.57 |
| | 4 | 2.53 ± 1.38 | 2.46 ± 1.36 | 4.16 ± 2.05 |
| | 6 | 3.05 ± 1.54 | 2.96 ± 1.53 | 5.08 ± 2.20 |
| | 8 | 3.62 ± 1.89 | 3.51 ± 1.88 | 6.17 ± 2.69 |
| | 10 | 3.78 ± 1.88 | 3.66 ± 1.87 | 6.48 ± 2.58 |
| Camera Position (m) | 0.1 | 0.08 ± 0.03 | 0.08 ± 0.03 | 0.10 ± 0.03 |
| | 0.2 | 0.15 ± 0.05 | 0.15 ± 0.05 | 0.20 ± 0.05 |
| | 0.4 | 0.31 ± 0.10 | 0.30 ± 0.10 | 0.40 ± 0.10 |
| | 0.6 | 0.47 ± 0.15 | 0.46 ± 0.16 | 0.61 ± 0.16 |
| | 0.8 | 0.63 ± 0.21 | 0.61 ± 0.21 | 0.81 ± 0.21 |
| | 1.0 | 0.77 ± 0.26 | 0.76 ± 0.26 | 1.00 ± 0.27 |
| Scene Scale ($k$) | 0.25 | 0.40 ± 0.21 | 0.39 ± 0.21 | 0.65 ± 0.29 |
| | 0.50 | 0.76 ± 0.42 | 0.73 ± 0.42 | 1.24 ± 0.59 |
| | 1.00 | 1.42 ± 0.83 | 1.38 ± 0.83 | 2.39 ± 1.19 |
| | 1.50 | 2.17 ± 1.23 | 2.10 ± 1.23 | 3.63 ± 1.73 |
| | 2.00 | 2.91 ± 1.74 | 2.81 ± 1.73 | 4.87 ± 2.45 |

Three error sources are examined independently: target detection error in pixels, camera position error in meters, and scene scaling factor $k$ with ± 3 px detection error and ± 0.2 m camera error. Each configuration is averaged over 500 Monte Carlo trials.

camera 2, we use the Efficient Perspective-n-Point algorithm (Supplementary Methods) to calculate the relative pose of camera 2 with respect to camera 0 (Fig. 2a, step 3). Based on the calculated poses of the three cameras relative to camera 0, we determine the coordinates of each camera in the camera 0 coordinate system. Considering the three cameras as three spatial points, we use their coordinates in the camera 0 coordinate system, along with their known coordinates in the world coordinate system, to calculate the similarity transformation matrix between these two coordinate systems (Fig. 2(a), step 4; Supplementary Methods).

After the phase transition, each frame in Phase II (Online Tracking) is processed in real time. For each incoming frame, the synchronized 2D detections from the three cameras are triangulated using the pre-computed camera poses. The computed 3D coordinates in the camera 0 coordinate system are then transformed to world coordinates using the similarity transformation matrix.

**Evaluation.** The metrics we report include RMSE, MAE, Maximum Error, and R-squared (see Methods). The results show that errors are almost negligible: the RMSE is $7.8 \times 10^{-3}$ m, the MAE is $7.6 \times 10^{-3}$ m, the Maximum Error is $8.5 \times 10^{-3}$ m, and the R-squared is almost 1. This demonstrates the precision of our theoretical method, showcasing the value of applied mathematics in providing precise calculations. Actually, the primary source of pure simulation error is from the approximation inherent in computer numerical solutions. A further offline theoretical validation is provided in Supplementary Note 5.

**Sensitivity analysis.** In engineering scenarios, various factors such as camera positioning errors, pixel detection deviations, and scene scaling

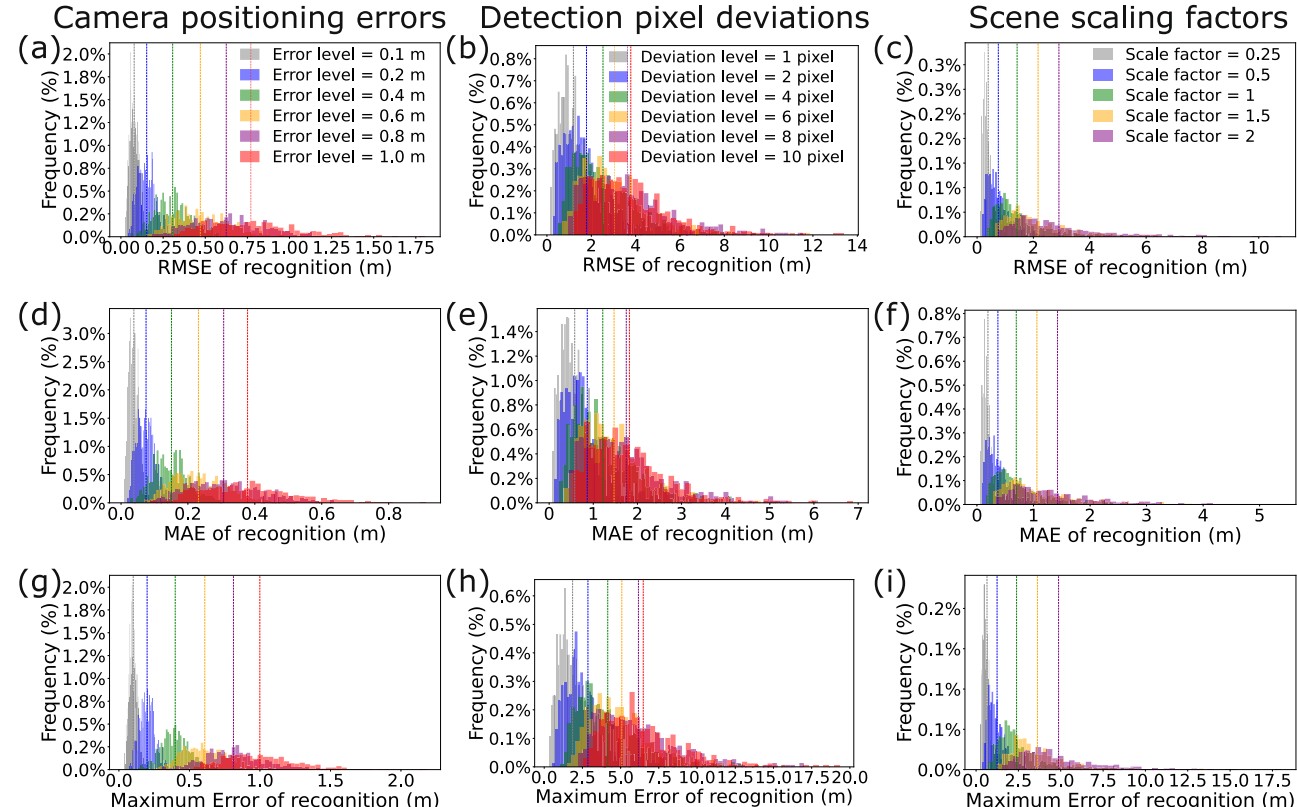

**Fig. 4 | Error histograms of dual-phase coordinate recognition under different perturbations.** For each perturbation level, 500 experiments are conducted within the dual-phase framework (Phase I: 300 frames for initialization; Phase II: 600 frames for online tracking). Histograms show the distribution of RMSE, MAE, and Maximum Error across trials, with dashed lines indicating mean values. **a–c** Camera positioning errors ($e_p$ = 0.1–1.0 m). **d–f** Pixel detection deviations ($e_{pix}$ = 1–10 pixels). **g–i** Scene scaling factors ($k$ = 0.25–2.0) with ± 0.2 m camera error and ± 3 pixel detection error.

**Fig. 5 | Schematic diagram of the UAV experiment for real-time UAV 3D coordinate recognition. a** Data Preprocessing: Collecting the images from the UAV flight captured by three cameras and dividing them into training and test sets with a ratio of 8:2. **b** YOLOv12 Model Training: Training the UAV detection model based on the YOLOv12 framework with a training set of UAV images captured in various scenes. **c** YOLOv12-Based UAV Prediction: The trained model is used to predict the bounding boxes of the UAV in the videos captured by the three cameras, and these raw per-frame outputs may still include missed and false detections. **d** YOLOv12-TS: The predicted UAV detections are refined using our proposed dual-phase YOLOv12-TS: Phase I applies both trajectory completion and outlier rejection; Phase II applies only outlier rejection. **e** Phase I: Batch Initialization: The refined 2D coordinate time series are used to estimate camera poses preparing for 3D coordinate recognition. **f** Phase II: Online Tracking: Real-time 3D coordinate recognition via SVD triangulation and similarity transform. The reconstructed trajectory is evaluated against ground-truth 3D coordinate data provided by the UAV onboard positioning device. The metrics used are RMSE, MAE, Maximum Error, and R-squared.

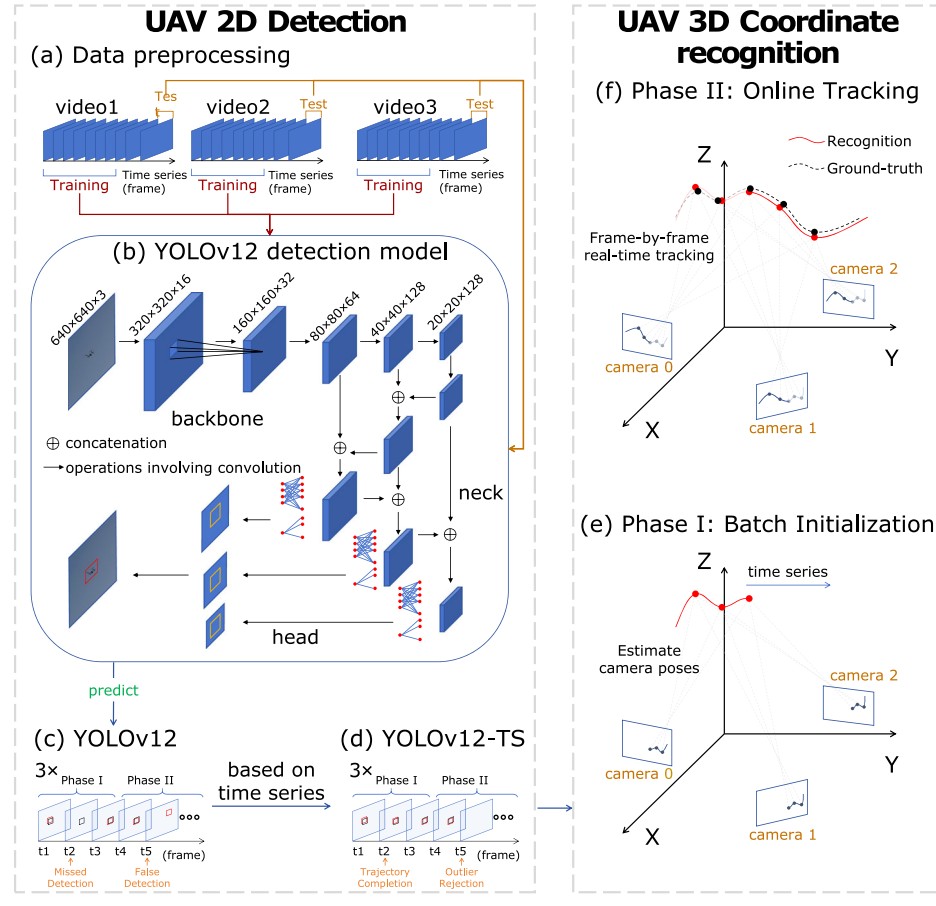

will impact the accuracy of 3D coordinate recognition. As shown in Fig. 4, to evaluate those factors we perform further simulations.

First, we explore the impact of camera positioning errors on the accuracy of 3D coordinate recognition by introducing random perturbations to each camera location. In particular, we consider six levels of positioning errors: $e_p = 0.1, 0.2, 0.4, 0.6, 0.8, 1.0$ m (Supplementary Note 1). For each level, the camera positions are perturbed by adding a randomly generated error vector to original positions:

$$G_{w,perturbed}^i = G_w^i + \mathbf{E}_p^i, \qquad (1)$$

where $G_w^i$ denotes the original position of camera $i$ in the world coordinate system, $G_{w,perturbed}^i$ represents the perturbed position, and $\mathbf{E}_p^i$ is a $3 \times 1$ vector in which each element is a random positioning error uniformly distributed within $[-e_p, e_p]$.

As shown in Table 1, for camera positioning errors of 0.1 m, 0.2 m, 0.4 m, 0.6 m, 0.8 m, and 1.0 m, the RMSE are 0.08 m, 0.15 m, 0.31 m, 0.47 m, 0.63 m, and 0.77 m, respectively, while the MAE are 0.08 m, 0.15 m, 0.30 m, 0.46 m, 0.61 m, and 0.76 m, respectively. The Maximum Error for these deviations are 0.10 m, 0.20 m, 0.40 m, 0.61 m, 0.81 m, and 1.00 m, respectively. As the camera positioning error increases, the error in 3D coordinate recognition also increases.

Second, we investigate the effect of 2D pixel coordinate deviations on the accuracy of 3D coordinate recognition. Let $e_{pix}$ represent the magnitude of pixel deviation. We consider six levels: $e_{pix} = 1, 2, 4, 6, 8, 10$ pixels. In each simulation, the 2D pixel coordinates of the spatial point in each image from the three cameras are perturbed by adding a randomly generated error:

$$O_{c'^i,perturbed}^j = O_{c'^i}^j + \mathbf{E}_{pix}^j, \qquad (2)$$

where $O_{c'^i}^j$ denotes the original 2D coordinates of the observed spatial point $O^j$ at time step $j$ in the pixel coordinate system $c'^i$, $O_{c'^i,perturbed}^j$ represents the perturbed 2D coordinates, and $\mathbf{E}_{pix}^j$ is a $2 \times 1$ vector representing the random pixel deviation uniformly distributed within $[-e_{pix}, e_{pix}]$.

For pixel deviations of 1, 2, 4, 6, 8, and 10 pixels (Table 1), the RMSE are 1.19 m, 1.78 m, 2.53 m, 3.05 m, 3.62 m, and 3.78 m, respectively, while the MAE are 1.17 m, 1.74 m, 2.46 m, 2.96 m, 3.51 m, and 3.66 m, respectively. The Maximum Error for these deviations are 1.86 m, 2.86 m, 4.16 m, 5.08 m, 6.17 m, and 6.48 m, respectively. As the pixel deviation increases, the error in 3D coordinate recognition also increases.

Third, we conduct a comprehensive study to assess the impact of scene scaling on the accuracy of 3D coordinate recognition, considering both camera positioning errors and pixel deviations. The scene is scaled using factors of $k = 0.25, 0.5, 1, 1.5, 2$. For each scaling factor, we scale the experimental scene while introducing a camera positioning error of 0.2 m and a random pixel deviation of 3 pixels. A scaling factor of 0.25 implies that the coordinates of both the spatial points and the camera positions are scaled to 25% of their original values.

Table 1 depicts the errors in the calculated 3D coordinates for each scaling factor, with RMSE of 0.40 m, 0.76 m, 1.42 m, 2.17 m, and 2.91 m, and corresponding MAE of 0.39 m, 0.73 m, 1.38 m, 2.10 m, and 2.81 m for scaling factors of 0.25, 0.5, 1, 1.5, and 2, respectively. The Maximum Error for these scaling factors are 0.65 m, 1.24 m, 2.39 m, 3.63 m, and 4.87 m, respectively. When both camera positioning errors and pixel deviations are considered, the larger the spatial scene, the greater the coordinate recognition errors.

**Coordinate recognition experiment: UAV case**
The UAV experiment is conducted at the First Stadium of Xianlin Campus, Nanjing University (118.952°E, 32.114°N). The experimental setup includes a UAV, three cameras, and a GNSS receiver (Supplementary Note 2). The

**Table 2 | Comprehensive benchmark of YOLO variants (v8-v12) with and without the Time Series (TS) module across three camera views**

| Dataset | Model | YOLO (%) | | | YOLO-TS (%) | | | Improvement (Δ, %) | | |
|---------|-------|-------|-------|-------|-------|-------|-------|--------|--------|--------|
| | | IoU-P | IoU-R | IoU-F1 | IoU-P | IoU-R | IoU-F1 | ΔIoU-P | ΔIoU-R | ΔIoU-F1 |
| Camera 1 | v12n | 97.70 | 99.44 | 98.56 | 98.86 | 99.69 | 99.27 | +1.16 | +0.25 | +0.71 |
| | v11n | 98.14 | 99.33 | 98.73 | 98.97 | 99.50 | 99.23 | +0.83 | +0.17 | +0.50 |
| | v10n | 98.98 | 98.62 | 98.80 | 99.69 | 98.87 | 99.28 | +0.71 | +0.25 | +0.48 |
| | v9t | 97.03 | 99.06 | 98.03 | 98.40 | 99.33 | 98.86 | +1.38 | +0.27 | +0.83 |
| | v8n | 97.57 | 99.37 | 98.46 | 98.82 | 99.62 | 99.22 | +1.25 | +0.25 | +0.76 |
| Camera 2 | v12n | 98.96 | 99.26 | 99.11 | 99.11 | 99.27 | 99.19 | +0.15 | +0.01 | +0.08 |
| | v11n | 99.20 | 99.52 | 99.36 | 99.32 | 99.53 | 99.43 | +0.13 | +0.01 | +0.07 |
| | v10n | 99.61 | 99.17 | 99.39 | 99.62 | 99.19 | 99.40 | +0.01 | +0.02 | +0.01 |
| | v9t | 98.75 | 99.47 | 99.11 | 98.95 | 99.47 | 99.21 | +0.20 | 0.00 | +0.10 |
| | v8n | 98.82 | 99.42 | 99.12 | 98.83 | 99.45 | 99.14 | +0.02 | +0.03 | +0.02 |
| Camera 3 | v12n | 98.21 | 98.21 | 98.21 | 98.31 | 98.21 | 98.26 | +0.10 | 0.00 | +0.05 |
| | v11n | 98.18 | 98.30 | 98.24 | 98.28 | 98.30 | 98.29 | +0.10 | 0.00 | +0.05 |
| | v10n | 98.47 | 97.73 | 98.10 | 98.49 | 97.73 | 98.11 | +0.02 | 0.00 | +0.01 |
| | v9t | 98.04 | 98.07 | 98.06 | 98.14 | 98.07 | 98.10 | +0.10 | 0.00 | +0.05 |
| | v8n | 97.92 | 98.17 | 98.04 | 97.97 | 98.17 | 98.07 | +0.06 | 0.00 | +0.03 |

The table details the IoU-Weighted Precision (IoU-P), Recall (IoU-R), and F1-score (IoU-F1). The improvements (Δ) indicate the performance gain achieved by the TS module.

process for UAV coordinate recognition in the world coordinate system is depicted in Fig. 5. To validate the proposed real-time hybrid framework under realistic engineering conditions, we process pre-recorded video streams sequentially to simulate online deployment. The framework comprises two functional modules: UAV detection, which acquires 2D coordinates as input; and UAV 3D coordinate recognition, the core of our study, which converts these 2D observations into 3D spatial coordinates. The two modules are applied under a dual-phase strategy: Batch Initialization, where observations are accumulated to refine detection accuracy and determine initial camera poses; and Online Tracking, where the system performs real-time 3D coordinate recognition with low latency. It is important to note that we consistently use the principles of time series to improve the accuracy of YOLOv12, the widely recognized top-performing UAV detection method based on CNNs, renaming it YOLOv12-TS (Time Series).

**AI-based UAV detection in time series images**. We conducted data collection using three ground-based cameras across three different scenes. From these videos, frames containing the UAV were extracted through frame-by-frame processing. To enhance UAV detection capabilities in complex sky backgrounds, 487 additional images were captured under scenarios such as building backgrounds, twilight, and backlighting (Supplementary Fig. 1). All images were manually annotated to ensure label quality. In total, a UAV dataset containing 48, 071 annotated images was constructed. An 8: 2 ratio was used to split the dataset into training and test sets, resulting in 38, 456 training images and 9 615 test images (Fig. 5a).

For UAV detection in time series images, YOLOv12n (You Only Look Once, version 12; Supplementary Note 3) is utilized as the detection framework (Fig. 5b) due to its lower computational cost and faster inference speed. The input resolution is modified from the default $640 \times 640 \times 3$ to $960 \times 960 \times 3$ to achieve finer detection granularity for small UAV targets. In addition, to verify the versatility of our framework, we conducted a comparative analysis using five YOLO versions (v8n, v9t, v10n, v11n, and v12n) trained with identical hyperparameters. The YOLO model is then utilized to detect UAV in time-series images captured by three cameras, providing basic detection results with bounding boxes for each image at every time step (Fig. 5c).

For each image, the model's predictions produce multiple bounding boxes with varying confidence levels. To filter potential UAV detections, a confidence threshold of 0.25 and an Intersection Over Union (IOU) threshold of 0.7 are applied, following the specifications in the YOLOv12 documentation. The confidence threshold sets the minimum confidence level required to accept detections, discarding those below this threshold. The IOU threshold is used for Non-Maximum Suppression to reduce duplicate detections. In this single-UAV scenario, only the detection with the highest confidence in each image is considered, even if multiple bounding boxes meet the thresholds. For the other four YOLO versions, we adopt the same hyperparameters in prediction.

When using the current state-of-the-art model YOLOv12 for UAV detection, challenges such as false detections caused by interference from other moving objects, like birds and dragonflies, and missed detections in complex backgrounds are encountered. To mitigate these issues, we develop YOLOv12-Time Series (YOLOv12-TS) (Fig. 5d), a refined version of YOLOv12 that incorporates time-series analysis and physics-informed speed modeling.

This approach leverages the distinct physical characteristics of the UAV, particularly its speed, which differs from other flying objects within the field of view, as well as the consistent motion state of the UAV over short time sequences. By strictly filtering targets based on these physical constraints, YOLOv12-TS effectively ensures that only physically valid observations are passed to the 3D coordinate solver. To balance detection quality with real-time requirements, YOLOv12-TS adopts a dual-phase processing strategy: during Batch Initialization, both trajectory completion and outlier rejection are applied to build a high-quality trajectory baseline for accurate pose estimation; during Online Tracking, only outlier rejection is performed for real-time operation.

We use IoU-P, IoU-R, and IoU-F1 as our evaluation metrics to assess the detection performance of YOLOv12 and YOLOv12-TS across three camera views, with the results presented in Table 2. Overall, YOLOv12-TS outperforms YOLOv12 across all metrics, with the most notable gains in Camera 1. The IoU-P increased from 97.70% to 98.86%, as the physics-informed gating mechanism removes false detections caused by dynamic environmental noise. Meanwhile, trajectory completion recovers missed detections, increasing IoU-R from 99.44 to 99.69%. The combined effect raises IoU-F1 from 98.56 to 99.27%.

This improvement is not limited to YOLOv12. Table 2 demonstrates that our proposed Time Series (TS) module consistently improves detection performance across all versions, confirming the method's generalizability

Fig. 6 | **Evaluation of UAV 3D coordinate recognition results with ground-truth along the X, Y, and Z axes.** The figure is divided into three sections: **a** shows the deviation of the UAV 3D coordinates along the X axis, **b** along the Y axis, and **c** along the Z axis. The solid red lines represent the UAV 3D coordinates obtained from the recognition process, while the dashed black lines indicate the corresponding ground-truth UAV 3D coordinates provided by the onboard positioning device.

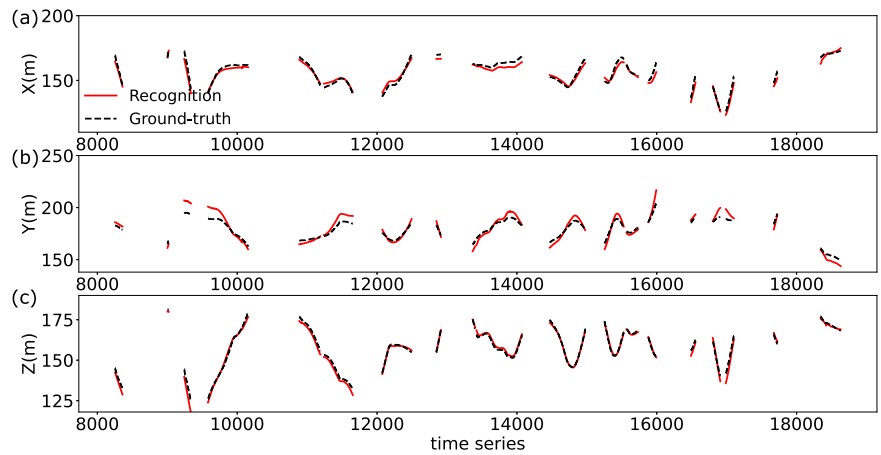

### Table 3 | Ablation study of BA backend on trajectory accuracy

| Model | Type | Overall | | | X-axis | | | Y-axis | | | Z-axis | | |
|---|---|---|---|---|---|---|---|---|---|---|---|---|---|
| | | RMSE | MAE | $R^2$ | RMSE | MAE | $R^2$ | RMSE | MAE | $R^2$ | RMSE | MAE | $R^2$ |
| YOLOv8 | w/o BA | 6.24 | 5.45 | 0.88 | 2.59 | 2.10 | 0.90 | 5.31 | 4.10 | 0.74 | 2.02 | 1.68 | 0.97 |
| | w/ BA | 5.66 | 4.96 | 0.91 | 2.71 | 2.18 | 0.90 | 4.65 | 3.65 | 0.81 | 1.76 | 1.30 | 0.98 |
| YOLOv9 | w/o BA | 6.00 | 5.13 | 0.89 | 2.69 | 2.15 | 0.90 | 5.14 | 3.92 | 0.74 | 1.53 | 1.21 | 0.99 |
| | w/ BA | 5.86 | 5.00 | 0.91 | 1.96 | 1.59 | 0.97 | 5.25 | 4.10 | 0.73 | 1.72 | 1.21 | 0.98 |
| YOLOv10 | w/o BA | 5.98 | 5.20 | 0.89 | 2.68 | 2.16 | 0.89 | 5.05 | 3.88 | 0.76 | 1.75 | 1.41 | 0.98 |
| | w/ BA | 5.60 | 4.84 | 0.91 | 2.19 | 1.87 | 0.95 | 4.80 | 3.77 | 0.79 | 1.85 | 1.36 | 0.98 |
| YOLO11 | w/o BA | 6.44 | 5.59 | 0.88 | 2.46 | 1.96 | 0.91 | 5.41 | 4.17 | 0.72 | 2.48 | 2.09 | 0.96 |
| | w/ BA | 6.33 | 5.43 | 0.89 | 2.23 | 1.91 | 0.96 | 5.42 | 4.26 | 0.73 | 2.39 | 1.72 | 0.96 |
| YOLOv12 | w/o BA | 6.30 | 5.48 | 0.87 | 2.93 | 2.44 | 0.87 | 5.29 | 4.13 | 0.73 | 1.76 | 1.39 | 0.98 |
| | w/ BA | 5.45 | 4.83 | 0.91 | 2.55 | 2.13 | 0.93 | 4.52 | 3.66 | 0.80 | 1.66 | 1.20 | 0.98 |

*RMSE* Root Mean Square Error (m), *MAE* Mean Absolute Error (m); $R^2$ R-squared (coefficient of determination), "*w/o BA*" without Bundle Adjustment (SVD-only), "*w/ BA*" with background Bundle Adjustment refinement.

across YOLO variants. Among these, YOLOv12n exhibits a favorable balance of inference speed and precision. We therefore adopt YOLOv12n as the primary detector for the remainder of this study.

**UAV 3D coordinate recognition.** Following the robust 2D detection provided by YOLOv12-TS, the 3D coordinate recognition module processes the synchronized 2D streams. To meet the real-time requirements, this module is executed according to the proposed dual-phase strategy.

In Phase I, after preprocessing for UAV detection results (as detailed in Supplementary Methods), we obtain synchronized and aligned UAV 2D coordinate time series from all three cameras under a unified time framework for UAV coordinate recognition (Fig. 5e). We first use the UAV 2D coordinate time series from camera 0 and camera 1 to calculate the relative poses between these two cameras. Specifically, we identify the common time period between these two time series and pair the UAV 2D coordinates at each time step. These coordinate pairs from all time steps are used to calculate the relative pose of camera 1 with respect to camera 0 (Fig. 2a, step 1).

Using the calculated relative poses of camera 0 and camera 1 along with the UAV 2D coordinate time series from these two cameras, we establish the geometric relationship and use SVD to obtain the UAV 3D coordinates in the camera 0 coordinate system for each time step (Fig. 2a, step 2).

For the remaining cameras i (where N > i≥2, with i = 2 and N = 3 in the UAV experiment), we use the preprocessed UAV 2D coordinate time series from camera i (Supplementary Methods) and the calculated 3D UAV coordinates in the camera 0 coordinate system to compute the relative pose of camera i with respect to camera 0 (Fig. 2(a), step 3). Additionally, the 2D coordinate time series from camera i provide supplementary observations

that are also used for UAV coordinate recognition (Fig. 2a, step 2; Supplementary Methods).

Based on the calculated poses of each camera relative to camera 0, we determine the coordinates of each camera within the camera 0 coordinate system. Considering the N cameras as N spatial points, we use their coordinates in the camera 0 coordinate system, along with their known coordinates in the world coordinate system obtained from the GNSS receiver, to calculate the similarity transformation matrix between the camera 0 coordinate system and the world coordinate system which is used for transforming the UAV 3D coordinate time series from the camera 0 coordinate system to the world coordinate system (Fig. 2a, step 4).

In Phase II, the system transitions to frame-by-frame processing, where real-time tracking is performed using our SVD-based method to convert the synchronized 2D detections from YOLOv12-TS into 3D world coordinates. Simultaneously, a sliding-window Bundle Adjustment (BA) module runs in the background at fixed intervals. It uses accumulated observations to refine camera poses and then dynamically updates the optimized parameters to the online tracking module. This design ensures that the trajectory remains consistent over extended flight durations without interrupting the real-time data stream.

To evaluate the results of UAV coordinate recognition (Fig. 5f), ground-truth UAV 3D coordinates are provided by an onboard GNSS receiver, which updates six times per second with a positioning accuracy of up to 1 m. Fig. 6 presents the UAV 3D coordinate recognition results along the X, Y, and Z axes in the world coordinate system. The metrics used in the UAV experiment include the overall and component-wise (X, Y, Z) RMSE, MAE, and R-squared (Table 3). For the overall recognition, the RMSE is

**Table 4 | Runtime performance of the simulation and UAV experiments**

| Metric | Simulation | UAV experiment |
|---|---|---|
| Initialization frames $N$ | 300 | 2000 |
| Phase I initialization time | $0.022 \pm 0.004$ s | $0.33 \pm 0.01$ s |
| Phase II per-frame latency | 0.012 ms | 0.039 ms |
| Real-time factor | 2882 | 865 |

All timing results are averaged over 10 repeated runs. The real-time factor is computed as the ratio of frame interval to per-frame latency at 30 FPS.

5.45 m, MAE is 4.83 m, and R-squared is 0.91. For the X-axis, the RMSE is 2.55 m, MAE is 2.13 m and R-squared is 0.93. For the Y-axis, the RMSE is 4.52 m, MAE is 3.66 m, and R-squared is 0.80. For the Z-axis, the RMSE is 1.66 m, MAE is 1.20 m, and R-squared is 0.98, making it the most accurate among the three axes. The results of the UAV experiment indicate that our proposed time-series-based applied mathematical method for spatial point coordinate recognition is both robust and effective, achieving an RMSE of 5.45 m and an R-squared of 0.91 in a $100 \times 100 \times 30$ m scenario, without prior knowledge of camera attitudes.

It is important to note that the flight videos selected for our test set were captured under adverse weather conditions, specifically on rainy days with substantially degraded lighting. Despite these challenges, the engineering results demonstrate a high level of performance, indicating the robustness of our approach. Furthermore, it should be emphasized that 2D object detection serves as the foundation for 3D trajectory estimation, and the majority of current research efforts are still focused on improving 2D detection accuracy. As shown in Table 2, our method outperforms the current state-of-the-art model, YOLOv12, in 2D detection. While there is currently no direct benchmark available for 3D trajectory estimation, the achieved R-squared score suggests that our results remain satisfactory even under such unfavorable environmental conditions.

**Ablation studies**. First, to evaluate the generalizability of the proposed Time Series (TS) module, we tested five YOLO variants: YOLOv8n, YOLOv9t, YOLOv10n, YOLO11n, and YOLOv12n. All models were trained and tested on the same dataset with identical hyperparameters.

As shown in Table 2, the TS module improves IoU-P, IoU-R, and IoU-F1 across all model versions and camera views. All improvements are positive, confirming that the TS module effectively enhances detection robustness regardless of the underlying CNN backbone.

Second, to evaluate the contribution of the BA backend, we compared the accuracy of 3D coordinate recognition with and without BA optimization across all five YOLO variants.

As detailed in Table 3, the integration of the BA backend consistently enhances the performance of 3D coordinate recognition across all model variants. Comparing configurations without BA refinement and with BA refinement reveals a universal reduction in the overall RMSE and MAE. For the primary YOLOv12 backbone, the backend optimization successfully reduced the overall RMSE from 6.30 to 5.45 m and improved the R-squared to 0.91. Notably, the Z-axis stability shows marked improvement in the optimized trajectory, confirming the backend's capability to mitigate drift over long-duration flights.

The results also confirm that the framework is model-agnostic: all tested YOLO versions produce valid trajectories. While earlier versions like YOLOv8 and YOLOv10 deliver competitive baseline performance, the 'YOLOv12 w/ BA' configuration achieves the lowest overall reconstruction error (5.45 m RMSE) and the highest trajectory correlation. Consequently, YOLOv12 was selected as the optimal frontend detector for the main experiments presented in the manuscript.

**Computational complexity**. We analyze the computational complexity of both phases and validate the runtime through 10 repeated experiments for the numerical simulation and UAV experiment.

In Phase I, $N$ frames are buffered and camera poses are computed using closed-form solutions: essential matrix decomposition via SVD, Efficient Perspective-n-Point for additional cameras, and Kabsch-based similarity transformation (Supplementary Methods). With RANSAC-based outlier rejection and a fixed number of cameras, the overall complexity is $\mathcal{O}(N)$. In the UAV experiment ($N = 2000$), initialization completes in $0.33 \pm 0.01$ s.

In Phase II, each incoming frame requires only fixed-size matrix operations, including two-view triangulation and coordinate transformation, yielding $\mathcal{O}(1)$ per-frame complexity. The measured latency is 0.039 ms per frame, which is approximately 850 times faster than the 30 FPS input rate with a frame interval of 33.3 ms, confirming that the system operates well within real-time constraints. A background Bundle Adjustment thread refines poses every 2000 frames without blocking the tracking loop.

As shown in Table 4, both scenarios achieve real-time factors exceeding 800. The timing differences primarily arise because Phase I processes $N$ accumulated frames, and the UAV experiment uses $N = 2000$, which is approximately 6.7 times that of the simulation's $N = 300$. Additionally, the UAV experiment operates on real camera imagery requiring detection inference and undistortion, whereas the simulation uses synthetic observations with known correspondences. Despite these differences, both scenarios process each frame more than 800 times faster than the input rate, demonstrating robust real-time capability.

## Discussion

Current methods for 3D coordinate recognition of non-cooperative targets in the world coordinate system typically depend on calibration objects or attitude measurement devices, which can be costly and impractical for widespread deployment. In contrast, we propose a real-time hybrid framework with a dual-phase strategy for 3D coordinate recognition that eliminates the need for attitude measurements, integrating AI-driven 2D detection technology with mathematically-based 2D-to-3D transformations.

The mathematical core of our framework utilizes time series analysis and SVD to calculate relative camera poses, providing an effective method for determining camera attitudes without external references. An SVD-based approach is employed to compute the similarity transformation matrix, enabling accurate camera-to-world coordinate transformations. This mathematical foundation ensures theoretically sound and computationally efficient results, laying a robust groundwork for geodetic positioning. Furthermore, the framework enhances AI-based detection by incorporating the physical characteristics of moving objects into detection algorithms, substantially improving the accuracy of 2D coordinate time series acquisition. This integration not only improves precision but also extends the applicability of our method to diverse scenarios, from tracking UAVs to monitoring wildlife such as birds, and even larger objects with distinct feature points. Crucially, this work carries broader implications. By seamlessly merging AI and applied mathematics, our framework offers a streamlined approach to geodetic practice, demonstrating versatility across multiple domains within Earth system science. For instance, while UAVs serve as a case study in our experiments, the methodology can be readily adapted to other aerial objects, offering enhanced observations for calculating camera poses and improving 3D coordinate recognition accuracy in multi-target scenarios.

Despite the mathematical rigor of our theoretical framework, real-world implementation introduces practical challenges. The accuracy of 3D coordinate recognition is primarily affected by 2D detection errors and camera positioning uncertainties, especially in large-scale environments. These deviations remain within acceptable bounds for most applications, and improvements can be achieved through more precise AI models and better sensor hardware. Our method nonetheless represents an advancement in vision-based geodetic positioning, offering a streamlined and scalable approach to 3D positioning without reliance on traditional attitude parameters. This work opens new possibilities for monitoring dynamic Earth systems using minimal sensing infrastructure.

## Methods

### Dual-phase strategy

To achieve real-time 3D coordinate recognition, we design a dual-phase strategy (Fig. 5; Algorithm 1). During Phase I, the system accumulates observations for detection refinement and pose estimation. Once the camera poses are determined, Phase II performs real-time coordinate recognition.

**Phase I: batch initialization.** The system buffers the initial $N$ frames from all cameras. The YOLOv12-TS module processes the buffered detections using both Outlier Rejection and Trajectory Completion, producing a refined 2D coordinate time series. This series is then passed to the geometric solver, which computes the camera poses and the similarity transform parameters. These parameters are then fixed for use in the subsequent phase.

**Phase II: online tracking.** After initialization, the system performs frame-by-frame processing for real-time coordinate recognition. For each incoming frame, YOLOv12-TS applies only Outlier Rejection to filter detections. The filtered 2D coordinates are then transformed to 3D world coordinates using the determined camera parameters.

In the UAV experiment, to further improve accuracy during extended flights, a backend module executes sliding-window refinement without blocking real-time processing. This module is triggered periodically, each time incorporating a longer observation history for camera pose estimation. Specifically, we employ Bundle Adjustment[26], a well-established numerical optimization technique in photogrammetry, to minimize reprojection errors. The refined parameters are then updated for Online Tracking.

**Algorithm 1**. Dual-Phase Real-Time 3D Coordinate Recognition

**Require:** Video stream frames $F_t^k$ from $K$ cameras, Buffer size $N$, Camera world coordinates $\{G^k\}_{k=1}^K$

**Ensure:** Real-time 3D world coordinates $P_w$

1: **Initialize:** $State \leftarrow$ INIT, Buffer $\mathcal{B}^k \leftarrow \varnothing$, $V^k \leftarrow 0$, $P_{last}^k \leftarrow null$ for each camera $k$
2: **for** each incoming frame $F_t$ at time $t$ **do**
3: **for** each camera $k = 1, \ldots, K$ **do**
4: $D_{raw}^k \leftarrow YOLOv12Inference(F_t^k)$
5: **end for**
6: **if** $State ==$ INIT **then**
7: *Phase I: Batch Initialization*
8: **for** each camera $k$ **do**
9: $\mathcal{B}^k.append(D_{raw}^k)$
10: **end for**
11: **if** $Length(\mathcal{B}^k) == N$ **then**
12: **for** each camera $k$ **do**
13: $\mathcal{B}^k \leftarrow OutlierRejection(\mathcal{B}^k)$
14: $\mathcal{B}^k \leftarrow TrajectoryCompletion(\mathcal{B}^k)$
15: **end for**
16: $\{R_{cw}^k, t_{cw}^k\} \leftarrow PoseEstimation(\{\mathcal{B}^k\}, \{G^k\})$
17: $\{s, R, t\} \leftarrow SimilarityTransform(\{\mathcal{B}^k\}, \{G^k\})$
18: $State \leftarrow$ ONLINE
19: **end if**
20: **Continue**
21: **else**
22: *Phase II: Online Tracking*
23: **for** each camera $k$ **do**
24: $P_{pred}^k \leftarrow P_{last}^k + V^k$
25: $Gate^k \leftarrow \lambda \cdot \|V^k\|$
26: **if** $D_{raw}^k$ is valid **and** $\| D_{raw}^k - P_{pred}^k \| < Gate^k$ **then**
27: $D_{out}^k \leftarrow D_{raw}^k$
28: $V^k \leftarrow D_{out}^k - P_{last}^k$
29: $P_{last}^k \leftarrow D_{out}^k$
30: **else**
31: **Skip** frame for camera $k$
32: **end if**
33: **end for**
34: $P_c \leftarrow Triangulation(\{D_{out}^k\}, \{R_{cw}^k, t_{cw}^k\})$
35: $P_w \leftarrow s \cdot R \cdot P_c + t$
36: **return** $P_w$
37: **end if**
38: **end for**

### Coordinate recognition framework

**Formulation of the coordinate system model.** Within a three-dimensional space, as shown in Fig. 2, we consider a set of ground-based cameras, each uniquely represented by its optical center for camera $i$, denoted as $G^i$, where $i = 0, 1, 2, \ldots, N - 1$, and an observed flying object denoted as $O^j$ at each time step $j$, where $j = 0, 1, 2, \ldots, M - 1$. The field of view of these cameras covers the object $O^j$ at all time steps.

The coordinates of a spatial point $P$ ($G^i$ or $O^j$) differ depending on the chosen coordinate system. In the trajectory reconstruction of flying object, the world coordinate system, camera coordinate system, and pixel coordinate system are typically employed. Here, the trajectory of the flying object is first obtained in the pixel coordinate systems of each camera, which is then transformed into the camera coordinate systems. Next, the trajectory in the camera coordinate systems can be converted to the world coordinate system via the similarity transformation. The schematic of our proposed methodology is shown in (Fig. 2).

World coordinate system: In this study, we use the World Geodetic System 1984[27] as the 3D Cartesian reference frame. As shown in (Fig. 2), our 3D Cartesian coordinate system has its origin at the Earth center of mass. We denote the world coordinate system as $w$. The coordinates of each point $P$ in $w$ are represented as $P_w = (X_w, Y_w, Z_w)$.

Camera coordinate system: This is also a 3D Cartesian coordinate system, with its origin located at the camera optical center in Fig. 2b. The Z-axis of the camera coordinate system is perpendicular to the camera imaging plane, while the X-axis and Y-axis are parallel to the horizontal and vertical directions of the imaging plane, respectively. We denote the camera coordinate system ($c$) for each camera $i$ as $c^i$. The coordinates of each point $P$ in $c^i$ are represented as $P_{c^i} = (X_{c^i}, Y_{c^i}, Z_{c^i})$.

Pixel coordinate system: This is a 2D coordinate system that represents the projected position of a point $P$ in 3D space onto the captured image (Fig. 2b). The origin of this coordinate system is located at the top-left corner of the image, with the $u$-axis pointing to the right and the $v$-axis pointing downwards. We denote the pixel coordinate system for each camera $i$ as $c'^i$. The coordinates of each point $P$ in $c'^i$ are $(u_{c'^i}, v_{c'^i})$. In this paper we use homogeneous coordinates $P_{c'^i} = (u_{c'^i}, v_{c'^i}, 1)$.

**Coordinate transformations.** Three types of coordinate transformations are involved in our framework.

3D to 3D: Based on a similarity transformation that includes a rotation matrix, a translation vector, and a scale factor, the coordinates of a point $P$ can be transformed from one 3D Cartesian coordinate systems $m$ to another $n$:

$$P_n = s_{nm}R_{nm}P_m + t_{nm}, \tag{3}$$

where $P_n$ denotes the coordinates of $P$ in the coordinate system $n$, $P_m$ denotes the coordinates of $P$ in the coordinate system $m$, $s_{nm}$ is a constant scale factor that represents the scaling from $m$ to $n$, $R_{nm}$ is a $3 \times 3$ rotation matrix from $m$ to $n$, and $t_{nm}$ is a $3 \times 1$ translation vector in $n$ pointing from the origin of $n$ to the origin of $m$. In particular, if the coordinate systems $n$ and $m$ are at the same scale, then $s_{nm} = 1$. For the world coordinate systems $w$ and the camera coordinate system $c$, we have:

$$P_c = R_{cw}P_w + t_{cw}. \tag{4}$$

3D to 2D: The point $P$ in the 3D camera coordinate system is transformed into the 2D pixel coordinate system based on the pinhole model (Fig. 2b), with this transformation process precisely described by the camera

intrinsic parameter matrix $K$:

$$K = \begin{bmatrix} f_x & 0 & d_x \\ 0 & f_y & d_y \\ 0 & 0 & 1 \end{bmatrix}, \quad (5)$$

where $f_x$ and $f_y$ are the focal lengths in the x and y directions, respectively, and $(d_x, d_y)$ is the principal point, which is the projection of the camera optical center onto the image plane[28].

Given a point $P = (X, Y, Z)$ in the camera coordinate system, the projected pixel coordinates $(u, v, 1)$ can be obtained by:

$$Z \begin{bmatrix} u \\ v \\ 1 \end{bmatrix} = K \begin{bmatrix} X \\ Y \\ Z \end{bmatrix} = \begin{bmatrix} f_x & 0 & d_x \\ 0 & f_y & d_y \\ 0 & 0 & 1 \end{bmatrix} \begin{bmatrix} X \\ Y \\ Z \end{bmatrix}. \quad (6)$$

2D to 3D: Eq. (6) maps a point $P$ from a 3D coordinate system to a 2D coordinate system, however this mapping is non-invertible due to the ambiguity in the depth dimension $Z$. Resolving depth $Z$ relies on identifying corresponding points across images from multiple cameras to establish spatial relationships.

Given the pixel coordinates $P_{c'^1}$ and $P_{c'^2}$ of point $P$ in the pixel coordinate systems $c'^1$ and $c'^2$, along with their rotation matrix $R_{c^2c^1}$ and translation vector $t_{c^2c^1}$, the 3D coordinates $P_{c^1}$ can be recovered through triangulation[21]:

$$P_{c^1} = \frac{-[P_{c'^2}]_\times K^2 R_{c^2c^1} K^{1-1} P_{c'^1} \cdot [P_{c'^2}]_\times K^2 t_{c^2c^1}}{[P_{c'^2}]_\times K^2 R_{c^2c^1} K^{1-1} P_{c'^1} \cdot [P_{c'^2}]_\times K^2 R_{c^2c^1} K^{1-1} P_{c'^1}} K^{1-1} P_{c'^1}, \quad (7)$$

denotes the skew-symmetric matrix of $P_{c'^2}$. The detailed derivation is provided in Supplementary Methods.

**3D coordinate recognition**. Given the world coordinates of multiple ground-based cameras, we implement the dual-phase strategy (Algorithm 1) as follows.

Batch Initialization: Given the 2D pixel coordinates of the observed object from two cameras, we first compute the fundamental matrix using epipolar geometry[21,29]. The essential matrix is then decomposed via SVD to extract the relative rotation and translation between cameras. Due to the inherent scale ambiguity in monocular reconstruction, we introduce a scaled coordinate system. Using triangulation, we compute the 3D coordinates of the observed object in this scaled system.

For additional cameras, we employ the Efficient Perspective-n-Point algorithm[30] to recover their poses from 2D to 3D correspondences[22]. Finally, to align the reconstruction with the world coordinate system, we apply the Kabsch algorithm[31] to compute the similarity transformation using the known camera positions in the world frame.

Online Tracking: Once the camera poses and transformation parameters are determined during initialization, real-time coordinate recognition reduces to a direct coordinate transformation. The world coordinate of the observed object at each time step is computed by applying the inverse similarity transformation to the triangulated 3D point. Complete mathematical derivations are provided in Supplementary Methods.

**YOLOv12-time series**

In YOLOv12-based UAV detection, we face two main challenges. First, other moving objects, such as birds, can cause interference. Second, missed detections can occur due to low confidence levels or complex backgrounds involving buildings. However, the temporal motion patterns of UAV flight are smooth and continuous. This property can be used to distinguish UAV from transient noise and to mitigate missed detections. Based on this, we propose YOLOv12-Time Series (YOLOv12-TS). As outlined in Algorithm 1, YOLOv12-TS employs a dual-phase strategy: during Batch Initialization,

the buffered sequence is processed as a whole to establish a reliable initial trajectory, filtering outliers and completing gaps; in Online Tracking, each incoming frame is processed immediately (Supplementary Note 4).

**Batch initialization**. The objective of this phase is to generate reliable 2D detection sequences for pose estimation. We buffer a sequence of frames and process them jointly using two operations.

Outlier Rejection: To filter out transient noise like birds, we verify kinematic consistency. For a detection at frame $i$ and a subsequent detection at frame $j$, if the Euclidean distance $d(i, j)$ satisfies $d(i, j) < v_{max} \times (j - i)$, where $v_{max}$ is a speed threshold derived using inter-quartile range filtering, the detection is physically accepted.

Trajectory Completion: To address missed detections that could affect pose estimation, we perform interpolation. If two confirmed detections at frames $i$ and $i + k$ are separated by a gap, we interpolate the missing coordinates assuming linear motion, yielding dense trajectory input.

**Online tracking**. Once the camera poses are determined, the system processes each incoming frame in real time. We implement a physics-informed gating mechanism for outlier rejection. Specifically, at each time step $t$, a dynamic search radius $r = \lambda \cdot v_{t-1}$ is defined around the predicted position $\hat{P}_t$, where $v_{t-1}$ is the velocity estimated from the previous frame and $\lambda$ is a scaling factor. A detection $D_t$ is accepted only if $\| D_t - \hat{P}_t \| < r$; otherwise, it is rejected to prevent drift.

**Metrics**

**Metrics for 2D object detection**. We evaluate the performance of 2D detection using Precision (IoU-P), Recall (IoU-R), and F1-score (IoU-F1) computed at different IoU thresholds. IoU-P measures the proportion of detected bounding boxes that correctly match ground-truth objects, IoU-R measures the proportion of ground-truth objects that are successfully detected, and IoU-F1 provides the harmonic mean of the two. These metrics assess the accuracy of predicted bounding boxes by measuring the overlap with ground-truth annotations across multiple IoU thresholds (Supplementary Methods).

**Metrics for 3D coordinate recognition**. We evaluate the performance of 3D coordinate recognition using four metrics: Root Mean Square Error (RMSE), Mean Absolute Error (MAE), Maximum Error, and R-squared. RMSE quantifies the overall deviation by penalizing larger errors more heavily, MAE provides an intuitive measure of average positional error, Maximum Error captures the worst-case deviation, and R-squared indicates how well the reconstructed trajectory explains the variance in the ground truth. Detailed definitions are provided in Supplementary Methods.

## Data availability

The UAV dataset used in this study is publicly available at https://github.com/wordbomb/PoseFree-GeoLocator.

## Code availability

The source code is publicly available at https://github.com/wordbomb/PoseFree-GeoLocator. The code was developed using Python with PyTorch v2.3.0 and CUDA 12.1. The benchmark YOLO detector is based on the Ultralytics framework (https://github.com/ultralytics/ultralytics).

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

## Acknowledgements

This work is supported by the National Natural Science Foundation of China (61803047), and the Social Sciences Fund of Jiangsu Province 24XWB004. Ke-ke Shang is supported by Jiangsu Qing Lan Project and the NJU-China Mobile Joint Research Institute. Michael Small is supported by the Australian Research Council Discovery Grant (DP200102961). Michael Small also acknowledges the support of the Australian Research Council through the Center for Transforming Maintenance through Data Science (IC180100030).

## Author contributions

J.Y. (Co-first author) was responsible for experiment design, linear algebra analysis, coding, data analysis, and drafting the manuscript. K.-k.S. (Co-first author & Corresponding author) contributed to analysis, experiment, and simulation design, supervision, and manuscript writing. M.S. provided guidance and assisted in manuscript reviewing.

## Competing interests

The authors declare no competing interests.
