## [Transparent Peer Review file · Communications Engineering]

Bridging Mathematical Modeling and AI for 3D Coordinate Recognition of Moving Objects without External Reference and Attitude Measurement

Corresponding Author: Professor Keke Shang

Version 0:

Reviewer comments:

Reviewer #1

(Remarks to the Author)

The paper presents a method for 3D coordinate recognition for uncooperating moving objects. The main idea relies on a deep learning method for target detection (YOLOv12) further improved to incorporate temporal constraints (YOLOv2-TS) and a mathematical model based on SVD derive relative camera poses and reconstruct 3D trajectories. Overall, the work is interesting and sufficiently novel. Experimental results are adequate at demonstrating the effectiveness of the method, although some improvements could be made. For example, the entire framework depends on how reliable 2D detection by YOLOv12 is. It is unclear how other object detection methods could affect the framework performance. The paper could also benefit from an expanded discussion on computational complexity of the proposed approach. Clarity of presentation could be enhanced by better explaining the setting of the problem upfront. Some typos and grammatical errors are also present (e.g. page 20 "haves", ...)

Reviewer #2

(Remarks to the Author)

The paper "proposes a novel framework for 3D coordinate recognition that eliminates the need for attitude measurements" (p.13, l.-12). The framework consists of two distinct components:

- 1) An object-detection pipeline based on YOLOv12 with a simple time-series post-processing (YOLOv12-TS) for obtaining 2D coordinates of a UAV across video frames;
- 2) A coordinate recognition pipeline, described by the authors as their "theoretical framework" (p.14, l.-10), which relies on standard multi-view geometry operations: estimation of the essential matrix via SVD, decomposition into rotation and translation, PnP pose recovery, and Kabsch alignment.

Results are presented on numerical simulations and three real-world experiments involving UAVs observed by three cameras.

Major Comments

1. Applicability of the object-detection pipeline

The proposed detection approach (Algorithm 1, p. 21) is inherently offline: the first loop iterates over the entire set of frames, meaning that future frames are required to confirm or interpolate detections in earlier ones. Consequently, the algorithm cannot operate in real time: at frame t , it must access frames $t + 1$, $t + 2$, ... and even the last frame (T ? M ? conflicting notation in the paper) to produce its output. This limitation severely undermines the claimed applicability to "real-world scenarios."

To make the method relevant for practical systems, the authors should redesign the algorithm as an online or causal procedure that estimates the UAV position at time t using only the current and past frames (for instance, by employing a temporal filter or predictive model).

2. Incremental contribution of the coordinate recognition framework and missing experimental comparison

The coordinate recognition part of the pipeline closely mirrors the standard multi-view reconstruction pipeline found in classical literature. Section 5.3's first three paragraphs correspond almost one-to-one to textbook derivations such as those in Simon J. D. Prince, *Computer Vision: Models, Learning, and Inference* (2012, Ch. 16.2–16.3) and Hartley & Zisserman, *Multiple View Geometry in Computer Vision* (2003, Ch. 11). As presented, the paper's "coordinate recognition" stage is therefore incremental at best, not a theoretical innovation.

If the authors claim novelty, they should rigorously differentiate their method from prior work and demonstrate measurable performance gains relative to, at least:

- Sturm & Triggs (1996), *Factorization Methods for Projective Structure and Motion*;
- Triggs et al. (2000), *Bundle Adjustment - A Modern Synthesis*.

Moreover, the algebraic derivations are not entirely clear. For example, in Eq. (25) the symbol $(O^{\{T\}}_{\{c^{\{0\}}\}})$ is not introduced, and its role in the equation is not explained.

Minor Comments

The repeated claim that the method is "error-free" (p.2, l.1; p.6, l.23; etc) is overstated. All numerical SVD-based reconstructions are affected by data noise, conditioning, and linearization errors; the paper should moderate this phrasing.

A discussion of runtime and computational complexity would help substantiate the claimed real-time applicability.

Overall Assessment

The paper's novelty and scientific contribution are limited. The detection component is not operationally viable in real-time settings, and the coordinate recognition pipeline largely reproduces established multi-view geometry methods without comparative validation.

The paper would benefit from:

- reformulating the detection method for online inference;
- providing detailed comparisons against classical SfM, factorization, and bundle-adjustment baselines; and
- clearly positioning the work within existing literature.

Recommendation: Reject.

Version 1:

Reviewer comments:

Reviewer #1

(Remarks to the Author)

The authors addressed all my concerns in a satisfactory manner. I have no further comments.

Reviewer #2

(Remarks to the Author)

Overall, I think the authors have successfully addressed the concerns previously raised. The main remaining issue, requiring a major revision, is now the presentation and organization of the manuscript: introducing a dual-phase / two-stage algorithm is a substantial change, but it currently reads more like an add-on than the result of a thorough rethinking of the method and its exposition.

More concretely, the paper should be reorganized so that the actual contribution is front-and-center. The authors themselves summarize the novelty as: "our novelty is in demonstrating that a carefully designed integration of AI detection and classical techniques can solve this previously unsolved problem setting:"

In that spirit, the main text should emphasize the integration - in particular the dual-phase strategy (Algorithm 1) - rather than spending substantial space on detailed derivations of standard formulas.

A few specific suggestions along these lines:

1) It would make more sense to move section "2 Numerical simulations of coordinate recognition" to the SM (Supplementary Material), since it pertains to the previous / non-online version of the algorithm. Conversely, move section "S7 Validation of Dual-Phase Strategy via Simulation" from the SM into the main paper, since it validates the revised core strategy.

2) Move section "5.3 Method of coordinate recognition" to SM as well. The equations there appear to be standard multi-view / geometric-method material; this section can be shortened in the main text to a high-level description with appropriate citations, with details deferred to an appendix.

3) Bring the dual-phase strategy (Algorithm 1 and the key supporting validation/ablation items currently in the SM: S8.1 and S8.2, S10) into the main text, and much earlier than the current placement (certainly not as late as ~page 21).

4) Section "5.6 Metrics" feels superfluous: the metrics listed are standard, so this section could be significantly shortened or moved to SM (or reduced to a brief statement with citations).

More generally, as it stands, too much of the manuscript's scientific "payload" appears to sit in the SM rather than in the main text, which is unusual and weakens the presentation.

Version 2:

Reviewer comments:

Reviewer #2

(Remarks to the Author)

I am satisfied with the revisions and have no additional comments.

Dear Editors,

We would like to thank you for your prompt processing of our manuscript and the reviewers for furnishing us with very helpful comments and suggestions. We are very pleased with, and encouraged by, their assessment. In the following we respond directly to the comments provided by the referee and describe the changes we have made to the manuscript.

Response to Reviewer 1

The paper presents a method for 3D coordinate recognition for uncooperating moving objects. The main idea relies on a deep learning method for target detection (YOLOv12) further improved to incorporate temporal constraints (YOLOv2-TS) and a mathematical model based on SVD derive relative camera poses and reconstruct 3D trajectories. Overall, the work is interesting and sufficiently novel. Experimental results are adequate at demonstrating the effectiveness of the method, although some improvements could be made.

Response: We appreciate your review of our manuscript and your valuable comments. We take your suggestions very seriously and have revised them accordingly.

Comment 1 For example, the entire framework depends on how reliable 2D detection by YOLOv12 is. It is unclear how other object detection methods could affect the framework performance.

Response: We sincerely thank the reviewer for the valuable comment. To address this concern, we conducted a comprehensive sensitivity analysis across five YOLO variants (YOLOv8n, YOLOv9t, YOLOv10n, YOLO11n, and YOLOv12n).

The selection of the YOLO family as the detection backbone is motivated by the hard real-time constraint of our system. Given the input frequency of 30 FPS, the total end-to-end latency must remain below 33 ms. Two-stage detectors (e.g., Faster R-CNN) and detection transformers (e.g., DETR) were excluded as they typically exceed this latency budget. Among single-stage detectors, YOLO represents the state-of-the-art framework that satisfies our real-time requirements.

Our experiments demonstrate that the proposed mathematical framework is not dependent on a specific detector version. As shown in the newly added Supplementary Table S2, even with the older YOLOv8n architecture ($mAP_{50-95} = 68.4\%$), the system successfully reconstructs valid 3D coordinates with RMSE = 5.66 m. As detection accuracy improves across versions (v8 \rightarrow v12), we observe a

corresponding decrease in positioning error, with YOLOv12n achieving the best performance (RMSE = 5.45 m). This confirms that our coordinate recognition method is robust across different detection backends while naturally benefiting from improved detector quality.

Changes:

- **UAV Experiment Section:** Added comparative analysis across five YOLO versions.

“To verify the versatility of our framework, we conducted a comparative analysis using five versions of YOLO architecture (v8n, v9t, v10n, v11n, and v12n). All models were trained on the identical dataset with unified hyperparameters (Supplementary Information Section S6).”

- **UAV Experiment Section (Model Evaluation):** Added detection performance comparison table and TS module generalization analysis.

“We use IoU-P, IoU-R, and IoU-F1 as our evaluation metrics to assess the detection performance of YOLOv12 and YOLOv12-TS across three camera views... Overall, YOLOv12-TS outperforms YOLOv12 across all metrics, with the most notable gains in Camera 1. The IoU-P increased from 97.70% to 98.86%, as the physics-informed gating mechanism removes false detections caused by dynamic environmental noise.”

“This improvement is not limited to YOLOv12. Experimental results (Supplementary Section S8.1) demonstrate that our proposed Time Series (TS) module consistently improves detection performance across all versions, confirming the method’s generalizability across YOLO variants.”

- **Prediction based on basic detection with YOLOv12:** Added note on unified hyperparameters.

“For the other four YOLO versions (v8n, v9t, v10n, and v11n), we adopt the same hyperparameters in prediction.”

- **Supplementary Information Section S6:** Added “AI Detector Selection and Sensitivity Analysis” with Table 3.

“The proposed architecture operates in a hard real-time environment designed to track uncooperative UAVs without external attitude measurements. Given the camera input frequency of 30 FPS, the inter-frame interval is approximately 33 ms. To ensure valid online tracking without frame buffering, the total end-to-end latency must remain below this 33 ms budget... As

shown in Table 3, the framework exhibits stability across all tested detectors. Even with the older YOLOv8n, the system produces valid 3D coordinates, albeit with higher error. As mAP_{50-95} improves from v8n to v12n, the 3D positioning error generally decreases.”

Table 3. Impact of Detector Accuracy on 3D Positioning Error. Comparison of YOLO variants (v8n–v12n). Model parameters, mAP_{50-95} , and latency are official benchmarks from Ultralytics. $RMSE_{3D}$ is the 3D positioning error from our UAV experiment. For per-camera detection metrics, see Supplementary Table 5.

Model	Key Architecture	Params (M)	mAP_{50-95} (%)	Latency (ms)	$RMSE_{3D}$ (m)
YOLOv8n	Anchor-Free / C2f	3.2	68.4	4.2	5.66
YOLOv9t	GELAN / PGI	2.0	70.1	5.1	5.86
YOLOv10n	NMS-Free	2.3	72.5	4.8	5.60
YOLO11n	C3k2 / C2PSA	2.6	74.8	5.3	6.33
YOLOv12n	FlashAttention	2.6	76.9	5.5	5.45

- **Supplementary Information Section 8.1:** Added “Generalization of the TS Module” with Table 5.

“To evaluate the generalizability of the proposed Time Series (TS) module, we tested five YOLO variants: YOLOv8n, YOLOv9t, YOLOv10n, YOLO11n, and YOLOv12n. All models were trained and tested on the same dataset (Cameras 1–3) with identical hyperparameters.”

“As shown in Table 5, the TS module improves Precision, Recall, and F1-score across all model versions and camera views. All improvements are positive, confirming that the TS module effectively enhances detection robustness regardless of the underlying CNN backbone.”

Table 5. Comprehensive benchmark of YOLO variants (v8–v12) with and without the Time Series (TS) module across three camera views. The table details the Precision (P), Recall (R), and F1-score. The improvements (Δ) indicate the performance gain achieved by the TS module.

Dataset	Model	YOLO (%)			YOLO-TS (%)			Improvement (Δ , %)		
		P	R	$F1$	P	R	$F1$	ΔP	ΔR	$\Delta F1$
Camera 1	v12n	97.70	99.44	98.56	98.86	99.69	99.27	+1.16	+0.25	+0.71
	v11n	98.14	99.33	98.73	98.97	99.50	99.23	+0.83	+0.17	+0.50
	v10n	98.98	98.62	98.80	99.69	98.87	99.28	+0.71	+0.25	+0.48
	v9t	97.03	99.06	98.03	98.40	99.33	98.86	+1.38	+0.27	+0.83
	v8n	97.57	99.37	98.46	98.82	99.62	99.22	+1.25	+0.25	+0.76
Camera 2	v12n	98.96	99.26	99.11	99.11	99.27	99.19	+0.15	+0.01	+0.08
	v11n	99.20	99.52	99.36	99.32	99.53	99.43	+0.13	+0.01	+0.07
	v10n	99.61	99.17	99.39	99.62	99.19	99.40	+0.01	+0.02	+0.01
	v9t	98.75	99.47	99.11	98.95	99.47	99.21	+0.20	0.00	+0.10
	v8n	98.82	99.42	99.12	98.83	99.45	99.14	+0.02	+0.03	+0.02
Camera 3	v12n	98.21	98.21	98.21	98.31	98.21	98.26	+0.10	0.00	+0.05
	v11n	98.18	98.30	98.24	98.28	98.30	98.29	+0.10	0.00	+0.05
	v10n	98.47	97.73	98.10	98.49	97.73	98.11	+0.02	0.00	+0.01
	v9t	98.04	98.07	98.06	98.14	98.07	98.10	+0.10	0.00	+0.05
	v8n	97.92	98.17	98.04	97.97	98.17	98.07	+0.06	0.00	+0.03

- **Supplementary Information Section S8.2:** Added “Ablation Study of BA Backend” with Table 6.

“To evaluate the contribution of the Bundle Adjustment (BA) backend, we compared the accuracy of coordinate recognition with and without BA optimization across all five YOLO variants.”

“As detailed in Table 6, the integration of the Bundle Adjustment (BA) backend consistently enhances tracking performance across all model variants. Comparing configurations without BA refinement (‘w/o BA’) and with BA refinement (‘w/ BA’) reveals a universal reduction in the Overall RMSE and MAE. For the primary YOLOv12 backbone, the backend optimization successfully reduced the Overall RMSE from 6.30 m to 5.45 m and improved the coefficient of determination (R^2) to 0.91. Notably, the Z-axis stability shows significant improvement in the optimized trajectory, confirming the backend’s capability to mitigate drift over long-duration flights.”

“The results also confirm that the framework is model-agnostic: all YOLO versions (v8–v12) produce valid trajectories. While earlier versions like YOLOv8 and YOLOv10 deliver competitive baseline performance, the ‘YOLOv12

w/ BA’ configuration achieves the lowest overall reconstruction error (5.45 m RMSE) and the highest trajectory correlation. Consequently, YOLOv12 was selected as the optimal frontend detector for the main experiments presented in the manuscript.”

Table 6. Ablation study of BA backend on trajectory accuracy. RMSE: Root Mean Square Error (m); MAE: Mean Absolute Error (m); R^2 : coefficient of determination. “w/o BA”: without Bundle Adjustment (SVD-only); “w/ BA”: with background Bundle Adjustment refinement.

Model	Type	Overall			X-axis			Y-axis			Z-axis		
		RMSE	MAE	R^2	RMSE	MAE	R^2	RMSE	MAE	R^2	RMSE	MAE	R^2
YOLOv8	w/o BA	6.24	5.45	0.88	2.59	2.10	0.90	5.31	4.10	0.74	2.02	1.68	0.97
	w/ BA	5.66	4.96	0.91	2.71	2.18	0.90	4.65	3.65	0.81	1.76	1.30	0.98
YOLOv9	w/o BA	6.00	5.13	0.89	2.69	2.15	0.90	5.14	3.92	0.74	1.53	1.21	0.99
	w/ BA	5.86	5.00	0.91	1.96	1.59	0.97	5.25	4.10	0.73	1.72	1.21	0.98
YOLOv10	w/o BA	5.98	5.20	0.89	2.68	2.16	0.89	5.05	3.88	0.76	1.75	1.41	0.98
	w/ BA	5.60	4.84	0.91	2.19	1.87	0.95	4.80	3.77	0.79	1.85	1.36	0.98
YOLO11	w/o BA	6.44	5.59	0.88	2.46	1.96	0.91	5.41	4.17	0.72	2.48	2.09	0.96
	w/ BA	6.33	5.43	0.89	2.23	1.91	0.96	5.42	4.26	0.73	2.39	1.72	0.96
YOLOv12	w/o BA	6.30	5.48	0.87	2.93	2.44	0.87	5.29	4.13	0.73	1.76	1.39	0.98
	w/ BA	5.45	4.83	0.91	2.55	2.13	0.93	4.52	3.66	0.80	1.66	1.20	0.98

Comment 2 The paper could also benefit from an expanded discussion on computational complexity of the proposed approach.

Response: We sincerely thank the reviewer for the constructive comment. We have added a comprehensive computational complexity analysis (Supplementary Section S9), including timing benchmarks based on 10 repeated experiments for both simulation and UAV scenarios.

Our system operates in two phases with distinct computational characteristics. In Phase I (Batch Initialization), the system buffers N frames and performs pose estimation using closed-form solutions: essential matrix decomposition via SVD, EPnP for additional camera poses, and similarity transformation. The complexity is $\mathcal{O}(N)$ for RANSAC-based estimation. With $N = 2000$ frames in the UAV experiment, this one-time initialization completes in 0.33 ± 0.01 s.

In Phase II (Online Tracking), each incoming frame requires only fixed-size matrix operations: two-view triangulation via SVD and similarity transformation. The per-frame complexity is $\mathcal{O}(1)$, resulting in an average latency of 0.039 ms (UAV experiment) and 0.012 ms (simulation). Given the input frequency of 30 FPS (frame interval ≈ 33.3 ms), Phase II achieves real-time factors of $865\times$ and $2882\times$ for the UAV and simulation scenarios, respectively, confirming that the system operates well within hard real-time constraints.

Additionally, the Bundle Adjustment backend runs in the background periodically (every 2000 frames) without blocking the real-time tracking loop. A detailed timing comparison across scenarios is provided in Supplementary Table 7.

Changes:

- **Discussion:** Added statement on computational efficiency.

*“This mathematical foundation ensures **theoretically sound and computationally efficient** results, laying a robust groundwork for geodetic positioning.”*

- **Supplementary Information Section S10:** Added “Computational Complexity Analysis” with detailed timing benchmarks (Table 7) for both simulation and UAV scenarios.

“Our system operates in two phases with distinct computational characteristics. To quantify performance, we conducted 10 repeated experiments for each scenario.”

*“**Phase I: Batch Initialization.** The system buffers N frames and performs pose estimation using closed-form solutions: essential matrix decomposition via SVD, EPnP for additional camera poses, and similarity transformation. The complexity is $\mathcal{O}(N)$ for RANSAC-based estimation with a fixed number of cameras. With $N = 2000$ frames in the UAV experiment, this one-time initialization completes in 0.33 ± 0.01 s (mean \pm std. dev. over 10 runs).”*

*“**Phase II: Online Tracking.** Each incoming frame requires only fixed-size matrix operations: two-view triangulation via SVD and similarity transformation. The per-frame complexity is $\mathcal{O}(1)$, resulting in an average latency of 0.039 ms per frame. Given the input frequency of 30 FPS (frame interval ≈ 33.3 ms), Phase II achieves a real-time factor of 865 \times , confirming that the system operates well within hard real-time constraints. The BA module runs in a separate thread every 2000 frames, adding no latency to the real-time tracking loop.”*

Table 7 Runtime performance comparison between simulation and UAV experiments.

Metric	Simulation	UAV Experiment
Initialization frames N	300	2000
Phase I initialization time	0.022 ± 0.004 s	0.33 ± 0.01 s
Phase II per-frame latency	0.012 ms	0.039 ms
Real-time factor (30 FPS)	2882 \times	865 \times

“As shown in Table 7, both scenarios achieve real-time factors exceeding $800\times$. The timing differences arise from two factors: (1) Phase I processes N accumulated frames, and the UAV experiment uses $N = 2000$ (approximately $6.7\times$ that of simulation’s $N = 300$), resulting in proportionally longer initialization; (2) The UAV experiment operates on real camera imagery requiring detection inference and undistortion, whereas simulation uses synthetic observations with known correspondences. Despite these differences, both scenarios achieve real-time factors exceeding $800\times$, demonstrating robust real-time capability across diverse operational conditions.”

Comment 3 Clarity of presentation could be enhanced by better explaining the setting of the problem upfront.

Response: We sincerely thank the reviewer for this constructive suggestion. We acknowledge that the problem setting was not sufficiently clear in the original manuscript and have revised the introduction accordingly.

Our problem setting involves real-time 3D coordinate recognition of airborne objects (demonstrated using a UAV) using multiple ground-based cameras without attitude measurement equipment. This scenario presents several distinct challenges:

(i) *Non-cooperative target.* The UAV is treated as a non-cooperative object, meaning we cannot directly access its onboard GNSS receiver for positioning. In our experiments, the UAV’s GNSS data is used solely as ground truth for validation purposes.

(ii) *Non-overlapping fields of view.* The cameras are installed at different locations with varying poses, resulting in almost no overlap in their fields of view. This makes conventional Structure-from-Motion approaches infeasible, as feature point matching between views fails entirely.

(iii) *Position-only camera information.* The cameras are equipped only with consumer-grade GNSS receivers, providing position information but no attitude (orientation) data. Expensive inertial measurement units (IMUs) or precise pointing devices are not available.

(iv) *Real-time constraint.* The positioning must be performed in real-time as the UAV flies, rather than through post-processing after data collection.

We have expanded the problem formulation in the revised manuscript to clearly articulate these challenges before presenting our solution.

Changes:

- **Introduction (Paragraph 2):** Clarified the limitation of existing methods for non-cooperative targets.

“These methods, however, are inapplicable to non-cooperative targets, such as unidentified Unmanned Aerial Vehicles (UAVs).”

“Passive methods are utilized for non-cooperative targets but typically rely on external calibration objects or expensive attitude measurement devices to determine poses.”

- **Introduction (Paragraph 3):** Expanded the problem setting with detailed explanation of why SfM is inapplicable and how our method addresses the gap.

“To avoid reliance on attitude measurement devices, vision-based methods such as Structure from Motion (SfM) have been developed to estimate camera poses and reconstruct 3D geometry directly from image sequences. However, SfM is fundamentally designed for scenarios with a moving camera observing a static scene, and requires rich environmental features for cross-view matching. In real-world scenarios, conventional SfM becomes ineffective when stationary cameras attempt to track a moving object against a featureless sky, as there are insufficient distinctive features for correspondence. Instead, we observe that the trajectory of the spatial target, captured as a position-time series across multiple cameras, can serve as a natural correspondence across views, effectively replacing the role of static scene features.”

“In this perspective, we propose an efficient and simplified optical measurement system that utilizes cameras to establish relative spatial relationships through captured 2D coordinate time series.”

“By integrating rapid AI-driven 2D detection with a computationally efficient Singular Value Decomposition (SVD) solver, our framework requires only the known 3D positions of cameras, rather than their absolute attitudes, to achieve real-time, reference-free 3D positioning even with consumer-grade devices.”

- **Figure 5 caption:** Revised to clarify the dual-phase workflow.

Fig. 5 Schematic diagram of the UAV experiment for **real-time UAV 3D coordinate recognition**. (a) Data Preprocessing: Collecting the images from the UAV flight captured by three cameras and dividing them into training and test sets with a ratio of 8:2. (b) YOLOv12 Model Training: Training the UAV detection model based on the YOLOv12 framework with a training set of UAV images captured in various scenes. (c) YOLOv12-Based UAV Prediction: The trained model is used to predict the bounding boxes of the UAV in the videos captured by the three cameras, and these raw per-frame outputs may still include missed and false detections. (d) YOLOv12-TS: The predicted UAV detections are refined using our proposed dual-phase YOLOv12-TS: Phase I applies both trajectory completion and outlier rejection; Phase II applies only outlier rejection (see Section 5.4 for details). (e) Phase I: Batch Initialization: The refined 2D coordinate time series are used to estimate camera poses preparing for 3D coordinate recognition (Section 5.3). (f) Phase II: Online Tracking: Real-time 3D coordinate recognition via SVD triangulation and similarity transform. The reconstructed trajectory is evaluated against ground-truth 3D coordinate data provided by the UAV onboard positioning device. The metrics used are RMSE, MAE, Maximum Error, and R-squared, as detailed in Section 5.6.

- **UAV Experiment Section:** Added dual-phase framework description.

“To validate the proposed real-time hybrid framework under realistic engineering conditions, we process pre-recorded video streams sequentially to simulate online deployment. The framework comprises two functional modules: UAV detection, which acquires 2D coordinates as input; and UAV 3D coordinate recognition, the core of our study, which converts these 2D observations into 3D spatial coordinates. The two modules are applied under a dual-phase strategy: Batch Initialization, where observations are accumulated to refine detection accuracy and determine initial camera poses; and Online Tracking, where the system performs real-time 3D coordinate recognition with low latency.”

Comment 4 Some typos and grammatical errors are also present (e.g. page 20 “haves”, ...)

Response: We thank the reviewer for pointing out these errors. We have carefully proofread the entire manuscript and corrected the identified typos and grammatical issues.

Changes:

- **Introduction:** Revised terminology for consistency.

*“Geodetic positioning emerged from ancient geometric problems and subsequently evolved into a core technology of geomatics... This evolution grants the technology systematic, dynamic, and interdisciplinary features, naturally **drawing on** insights from physics, mathematics, with a growing emphasis on artificial intelligence.”*

- **Discussion:** Corrected phrasing and improved clarity.

*“Current methods for 3D **coordinate recognition** of non-cooperative targets in the world coordinate system typically depend on calibration objects or attitude measurement devices... In contrast, we propose a **real-time hybrid** framework for 3D coordinate recognition that eliminates the need for attitude measurements.”*

*“The accuracy of 3D coordinate **recognition** is primarily affected by 2D detection errors and camera positioning uncertainties.”*

*“Our method nonetheless represents a significant advancement in **vision-based geodetic positioning**, offering a streamlined and scalable approach to 3D positioning.”*

- **Methods Section:** Corrected “haves” → “have” and other grammatical is-

sues.

- **Throughout manuscript:** Proofread and corrected additional grammatical issues, including terminology standardization (e.g., “localization” → “coordinate recognition”) and sentence structure improvements.

Response to Reviewer 2

The paper “proposes a novel framework for 3D coordinate recognition that eliminates the need for attitude measurements” (p.13, l.-12). The framework consists of two distinct components: 1) An object-detection pipeline based on YOLOv12 with a simple time-series post-processing (YOLOv12-TS) for obtaining 2D coordinates of a UAV across video frames; 2) A coordinate recognition pipeline, described by the authors as their “theoretical framework” (p.14, l.-10), which relies on standard multi-view geometry operations: estimation of the essential matrix via SVD, decomposition into rotation and translation, PnP pose recovery, and Kabsch alignment.

Results are presented on numerical simulations and three real-world experiments involving UAVs observed by three cameras.

Response: We appreciate your review of our manuscript and your valuable suggestions; your feedback is essential for improving the quality of our research. We have further refined the presentation of the paper to highlight the contributions of the study more clearly. Below are our responses to specific comments.

Major Comments

1. Applicability of the object-detection pipeline

Comment 1 The proposed detection approach (Algorithm 1, p. 21) is inherently offline: the first loop iterates over the entire set of frames, meaning that future frames are required to confirm or interpolate detections in earlier ones. Consequently, the algorithm cannot operate in real time: at frame t , it must access frames $t + 1$, $t + 2$, ... and even the last frame (T ? M ? conflicting notation in the paper) to produce its output. This limitation severely undermines the claimed applicability to “real-world scenarios.”

To make the method relevant for practical systems, the authors should redesign the algorithm as an online or causal procedure that estimates the UAV position at time t using only the current and past frames (for instance, by employing a temporal filter or predictive model).

Response: We sincerely thank the reviewer for this critical observation, which has led to a substantial improvement of our work. Following this suggestion, we have completely redesigned the system architecture into a **dual-phase strategy** that is explicitly causal and suitable for real-time deployment.

In Phase I (Batch Initialization), the system buffers the first N_w frames to establish a robust foundation. During this phase, YOLO-TS performs outlier rejection and trajectory completion using the full warmup sequence, while the 3D coordinate recognition module estimates camera attitudes. This initialization phase is inherently offline but executes only once at system startup.

In Phase II (Online Tracking), the system transitions to a fully streaming, causal pipeline. Each incoming frame is processed by YOLO for object detection, followed by YOLO-TS performing only outlier rejection (without requiring future frames). Using the camera attitudes obtained from Phase I, the system performs real-time 3D coordinate recognition with a per-frame latency of only 0.039 ms.

Additionally, we incorporate a backend refinement strategy during online tracking. As more observations accumulate over time, the system periodically re-estimates camera attitudes using all historical data and refines them via bundle adjustment. These updated attitudes are then fed back to the online tracking module without interrupting the real-time operation.

This redesigned architecture ensures that at any time t , the system uses only current and past observations, fully addressing the reviewer’s concern about causality.

Changes:

- **Algorithm 1:** Completely rewritten as a dual-phase causal algorithm (YOLOv12-TS Dual-Phase Strategy).

“Require: Video Stream Frame F_t , Initialization Buffer Size N ”

“Ensure: Refined Detection D_{out} ”

“Initialize: State \leftarrow INIT, Buffer $\mathcal{B} \leftarrow \emptyset$ ”

*“Phase I: Batch Initialization. For each incoming frame, append D_{raw} to buffer \mathcal{B} . When $\text{Length}(\mathcal{B})$ equals N : apply *OutlierRejection* and *TrajectoryCompletion*; output \mathcal{B} for pose estimation; set State to ONLINE.”*

“Phase II: Online Tracking. Compute predicted position and gate threshold based on velocity. If detection is valid and within gate: accept detection; otherwise: skip frame.”

- **Methods Section (YOLOv12-TS):** Added detailed dual-phase description.

“YOLOv12-TS employs a dual-phase strategy: during Batch Initialization, the buffered sequence is processed as a whole to establish a reliable initial trajectory, filtering outliers and completing gaps; in Online Tracking, each incoming frame is processed immediately.”

*“**Batch Initialization.** The objective of this phase is to generate reliable 2D detection sequences for pose estimation. We buffer a sequence of frames and process them jointly using two operations: **Outlier Rejection** (to filter out transient noise like birds, we verify kinematic consistency using speed threshold derived from IQR filtering) and **Trajectory Completion** (to address missed detections, we perform interpolation assuming linear motion).”*

*“**Online Tracking.** Once the camera poses are determined, the system processes each incoming frame in real time. We implement a physics-informed gating mechanism for outlier rejection. At each time step t , a dynamic search radius $r = \lambda \cdot v_{t-1}$ is defined around the predicted position \hat{P}_t . A detection D_t is accepted only if $\|D_t - \hat{P}_t\| < r$; otherwise, it is rejected to prevent drift.”*

- **UAV Experiment Section:** Added dual-phase framework description.

“To validate the proposed real-time hybrid framework under realistic engineering conditions, we process pre-recorded video streams sequentially to simulate online deployment. The framework comprises two functional modules: UAV detection, which acquires 2D coordinates as input; and UAV 3D coordinate recognition, the core of our study, which converts these 2D observations into 3D spatial coordinates. The two modules are applied under a dual-phase strategy: Batch Initialization, where observations are accumulated to refine detection accuracy and determine initial camera poses; and Online Tracking, where the system performs real-time 3D coordinate recognition with low latency.”

*“**Phase II: Online Tracking.** Once the camera poses are determined in Phase I, the system transitions to frame-by-frame processing. In this phase, real-time tracking is performed using our linear SVD-based method. This approach guarantees ultra-low latency, converting the synchronized 2D detections from YOLOv12-TS into 3D world coordinates. Simultaneously, a sliding-window Bundle Adjustment (BA) module runs in the background at fixed intervals. It uses accumulated observations to refine camera poses and then dynamically updates the optimized parameters to the online tracking module.”*

- **Supplementary Information Section S7:** Added Section S7 “Dual-Phase Validation” with numerical simulation results.

“Building upon the numerical simulation framework, this section validates the dual-phase strategy under realistic streaming conditions. The trajectory is divided into two phases: During Phase I, the first $N = 300$ frames are buffered for Batch Initialization. After the phase transition, each frame in Phase II is processed in real time. Under ideal detection conditions, the pipeline achieves sub-centimeter accuracy (RMSE = 7.8 mm, Max = 8.5 mm) over 600 online frames with negligible latency (mean 0.021 ms per frame).”

2. Incremental contribution of the coordinate recognition framework and missing experimental comparison

Comment 2 The coordinate recognition part of the pipeline closely mirrors the standard multi-view reconstruction pipeline found in classical literature. Section 5.3’s first three paragraphs correspond almost one-to-one to textbook derivations such as those in Simon J. D. Prince, Computer Vision: Models, Learning, and Inference (2012, Ch. 16.2–16.3) and Hartley & Zisserman, Multiple View Geometry in Computer Vision (2003, Ch. 11). As presented, the paper’s “coordinate recognition” stage is therefore incremental at best, not a theoretical innovation.

If the authors claim novelty, they should rigorously differentiate their method from prior work and demonstrate measurable performance gains relative to, at least: - Sturm & Triggs (1996), Factorization Methods for Projective Structure and Motion; - Triggs et al. (2000), Bundle Adjustment - A Modern Synthesis.

Response: We thank the reviewer for this important observation and the opportunity to clarify our contribution. This feedback has prompted us to further refine our claims and improve our presentation. We fully acknowledge that the individual components of our coordinate recognition pipeline (essential matrix estimation, PnP, bundle adjustment) are well-established techniques from the multi-view geometry literature.

The core contribution of our work lies in the **novel integration of AI-driven detection with classical geometric methods** to address a previously unsolved problem. By combining rapid 2D object detection via deep learning with SVD-based efficient geometric solvers, we achieve real-time 3D coordinate recognition without attitude measurement equipment. This integration overcomes three key challenges simultaneously: (i) real-time computational constraints; (ii) robust and fast target detection from complex backgrounds via AI; (iii) camera attitude estimation without overlapping texture features across views.

Regarding comparison with prior methods, **we have conducted a systematic analysis:**

Structure-from-Motion (SfM). Classical SfM methods, including those of Sturm & Triggs (1996), fundamentally rely on feature point correspondences across overlapping views. In our scenario, the ground-based cameras observe the same airborne target but share no common background features, making traditional feature matching entirely infeasible. We have documented this limitation in Supplementary Information 5.

Bundle Adjustment (BA). BA, as synthesized by Triggs et al. (2000), is a powerful numerical optimization technique widely used in photogrammetry. However, BA requires good initial estimates to converge and is computationally expensive. In our improved framework, we leverage BA not as a standalone solution but as an asynchronous refinement module. During online tracking, BA runs periodically in the background using accumulated observations to refine camera attitudes, without interrupting the real-time stream. This design allows us to benefit from BA's optimization capability while maintaining real-time performance.

In summary, **our novelty is in demonstrating that a carefully designed integration of AI detection and classical techniques can solve this previously unsolved problem setting.** We also sincerely thank the reviewers for their insightful comments. In particular, their suggestions regarding Structure-from-Motion (SfM) and Bundle Adjustment (BA) significantly improved the mathematical optimization component of our paper, thereby enhancing its overall competitiveness.

Changes:

- **Introduction:** Clarified the contribution.

“By integrating rapid AI-driven 2D detection with a computationally efficient Singular Value Decomposition (SVD) solver, our framework requires only the known 3D positions of cameras, rather than their absolute attitudes, to achieve real-time, reference-free 3D positioning even with consumer-grade devices.”

- **Discussion:** Added paragraph positioning our work.

“The mathematical core of our framework utilizes time series analysis and Singular Value Decomposition (SVD) to calculate relative camera poses, providing an effective method for determining camera attitudes without external references.”

- **Supplementary Information Section S5:** Added “Applicability Analysis of Conventional Methods” with Table 2 comparing our method against classical SfM and BA-based approaches.

“Standard multi-view geometry approaches, specifically Structure from Motion (SfM) and Global Bundle Adjustment (BA), are widely used in 3D reconstruction. However, our experimental setup presents a degenerate case for these classical methods: three stationary ground-based cameras tracking a dynamic UAV against a featureless sky background.”

Table 2 Comparison of Multi-view Geometry Approaches. Comparison of conventional methods with our proposed hybrid framework under real-time and featureless constraints.

Aspect	Feature-based SfM	Global BA	Ours
Input Data	Keypoints	Initial Pose Guess	2D Detection
Initialization	Fail	Fail	SVD (closed-form)
Per-frame Complexity	$O(N^2)$	$O(N^3)$	$O(1)$
Featureless Scene	Fail	Requires prior	Works
Real-time	Fail	Fail	Works

*“**Baseline A: Failure of Conventional Feature-based SfM.** Standard SfM is inapplicable to our problem due to a fundamental inversion of the observation model: SfM assumes a moving camera observing a static scene, whereas our scenario involves stationary cameras observing a dynamic object. Three specific issues arise:”*

*“**1. Feature Starvation on Textureless Targets:** SfM relies on local descriptors (SIFT/ORB) to find correspondences. In our setting, texture extraction from the target is difficult due to its small size, and the background is dominated by sky. In our experiments, the number of detectable feature points on the UAV surface was frequently zero or insufficient for epipolar geometry estimation.”*

*“**2. Rejection of the Dynamic Target:** SfM assumes that the scene is static. In our scenario, the UAV is the only moving element. During geometric verification, algorithms classify moving target features as outliers to preserve background consistency.”*

*“**3. The Stationary Camera Configuration:** Our cameras are fixed on the ground. Classical SfM relies on camera motion to recover 3D structure from video sequences. Since our cameras do not move, no geometric parallax is generated over time.”*

“Baseline B: Infeasibility of Pure Global Bundle Adjustment. Global BA requires high-quality initialization (the Initialization Barrier) and violates the hard real-time requirement due to $O(N^3)$ complexity (the Latency Barrier).”

- **Supplementary Figure S5:** Added visualization of SfM feature matching failure.

Fig. S5. Visualization of SfM Feature Matching Failure. (a) and (b) represent two frames from the stationary camera sequence. Red circles indicate the ground-truth position of the target UAV. The colored lines represent computed feature matches. Note the phenomenon of “Feature Starvation.” The geometric feature matcher fails to identify any valid keypoints within the UAV region (marked by red circles). Matches are exclusively concentrated on the static background (trees and buildings), and many are mismatched (crossing lines) due to the non-rigid nature of the vegetation. The sky region remains completely devoid of features. Consequently, the SfM solver cannot recover the UAV’s motion.

“Visualization of SfM Feature Matching Failure. (a) and (b) represent two frames from the stationary camera sequence. Red circles indicate the ground-truth position of the target UAV. The colored lines represent computed feature matches. Note the phenomenon of “Feature Starvation.” The geometric feature matcher fails to identify any valid keypoints within the UAV region. Matches are exclusively concentrated on the static background (trees and buildings), and many are mismatched (crossing lines) due to the non-rigid nature of the vegetation. The sky region remains completely devoid of features.”

- **Strategic Solution:** Added explanation of our dual-phase design as a response to BA limitations.

“Strategic Solution: Dual-Phase Strategy with Background Refinement. To combine the high precision of BA with strict real-time constraints, we designed a dual-phase strategy that decouples refinement from real-time tracking. During Phase II (Online Tracking), the system uses SVD-based triangulation, which operates in constant $O(1)$ time per frame. This guarantees low

latency and provides initial pose estimates that bypass the initialization barrier. The Background Refinement Module executes sliding-window Bundle Adjustment at fixed intervals without blocking real-time processing.”

Comment 3 Moreover, the algebraic derivations are not entirely clear. For example, in Eq. (25) the symbol ($O_{c^0}^T$) is not introduced, and its role in the equation is not explained

Response: We sincerely thank the reviewer for this valuable comment. We have revised Eq. (25) to improve its clarity and consistency with Eq. (24).

In the original manuscript, Eq. (25) used a different notation style from Eq. (24), which caused confusion. In the revised version, we maintain the same dot product notation throughout to ensure consistency:

$$O_{\hat{c}^0}^j = - \frac{[O_{c^1}]_{\times} K^1 \tilde{R}_{c^1 c^0} K^{0^{-1}} O_{c^0} \cdot [O_{c^1}]_{\times} K^1 \tilde{t}_{c^1 c^0}}{[O_{c^1}]_{\times} K^1 \tilde{R}_{c^1 c^0} K^{0^{-1}} O_{c^0} \cdot [O_{c^1}]_{\times} K^1 \tilde{R}_{c^1 c^0} K^{0^{-1}} O_{c^0}} K^{0^{-1}} O_{c^0}^j. \quad (25)$$

We have also carefully reviewed all equations and notations throughout the manuscript to ensure clarity and that all symbols are properly introduced before use.

Changes:

- **Methods Section** (Eq. 25): Revised equation notation to use consistent dot product format matching Eq. 24.

$$“O_{\hat{c}^0}^j = - \frac{[O_{c^1}]_{\times} K^1 \tilde{R}_{c^1 c^0} K^{0^{-1}} O_{c^0} \cdot [O_{c^1}]_{\times} K^1 \tilde{t}_{c^1 c^0}}{[O_{c^1}]_{\times} K^1 \tilde{R}_{c^1 c^0} K^{0^{-1}} O_{c^0} \cdot [O_{c^1}]_{\times} K^1 \tilde{R}_{c^1 c^0} K^{0^{-1}} O_{c^0}} K^{0^{-1}} O_{c^0}^j ”$$

- **Methods Section:** Reviewed and clarified all symbol definitions.

Minor Comments

Comment 4 The repeated claim that the method is ”error-free” (p.2, l.1; p.6, l 23; etc) is overstated. All numerical SVD-based reconstructions are affected by data noise, conditioning, and linearization errors; the paper should moderate this phrasing.

Response: We thank the reviewer for pointing out this imprecise language. It is absolutely correct that all numerical methods, including SVD-based reconstructions, are subject to data noise, numerical conditioning, and linearization errors. Our original use of “error-free” was intended to describe the theoretical property under ideal noise-free conditions (as demonstrated in our numerical simulations),

but this phrasing is indeed misleading in practical contexts. We have revised all instances of “error-free” throughout the manuscript to more accurate expressions such as “closed-form solution” or “direct algebraic solution,” which correctly convey the deterministic nature of our method without implying immunity to real-world errors.

Changes:

- **Abstract:** Completely revised to remove “error-free” claim and clarify contribution.

“We propose a real-time hybrid framework for 3D coordinate recognition that integrates AI-driven detection with mathematical modeling. By utilizing time series, the nature of dynamic objects, our approach employs computationally efficient Singular Value Decomposition (SVD) to determine relative attitudes and achieve 3D coordinate recognition without absolute measurements.”

*“We enhance the state-of-the-art You Only Look Once version 12 (YOLOv12) model by incorporating time-series analysis for **rapid and precise** 2D detection, which serves as input for 2D-to-3D conversion via our **SVD-based solver**.”*

“This fusion of AI and applied mathematics provides a streamlined approach for real-time 3D coordinate recognition, eliminating the reliance on traditional attitude measurement.”

- **Numerical Simulations Section:** Revised precision description to accurately describe error sources.

*“All in all, our simulations show that using applied mathematics to locate spatial points **achieves high precision; residual errors in pure simulation stem solely from finite floating-point precision. When equipment errors are introduced, the errors we observe stem from these input inaccuracies: the higher the precision of the equipment, the more accurate the spatial point coordinate recognition. Even with equipment errors, the applied mathematical framework remains robust and effective.**”*

- **Throughout manuscript:** Replaced “error-free” with “closed-form solution,” “direct algebraic solution,” or “deterministic computation.”

Comment 5 A discussion of runtime and computational complexity would help substantiate the claimed real-time applicability.

Response: We thank the reviewer for this suggestion. We have added a comprehensive computational complexity analysis to the revised manuscript (Supplementary Section S10), including timing benchmarks based on 10 repeated experiments for both simulation and UAV scenarios.

Our system operates in two phases with distinct computational characteristics. In Phase I (Batch Initialization), the system buffers N frames and performs pose estimation using closed-form solutions with complexity $\mathcal{O}(N)$. With $N = 2000$ frames in the UAV experiment, this one-time initialization completes in 0.33 ± 0.01 s.

In Phase II (Online Tracking), each frame requires only fixed-size matrix operations: two-view triangulation and similarity transformation. The per-frame complexity is $\mathcal{O}(1)$, resulting in an average latency of 0.039 ms (UAV experiment) and 0.012 ms (simulation). Given the input frequency of 30 FPS (frame interval ≈ 33.3 ms), Phase II achieves real-time factors of $865\times$ and $2882\times$ for the UAV and simulation scenarios, respectively, confirming that the system operates well within hard real-time constraints.

A detailed timing comparison is provided in Supplementary Table 7.

Changes:

- **Discussion:** Added statement on real-time capability.

*“This mathematical foundation ensures **theoretically sound and computationally efficient** results, laying a robust groundwork for geodetic positioning.”*

- **UAV Experiment:** Clarified the real-time processing capability.

*“**Phase II: Online Tracking.** Once the camera poses are determined in Phase I, the system transitions to frame-by-frame processing. In this phase, real-time tracking is performed using our linear SVD-based method. This approach guarantees ultra-low latency, converting the synchronized 2D detections from YOLOv12-TS into 3D world coordinates.”*

- **Supplementary Information Section S10:** Added “Computational Complexity Analysis” with Table 7 providing detailed timing benchmarks based on 10 repeated experiments.

“Our system operates in two phases with distinct computational characteristics. To quantify performance, we conducted 10 repeated experiments for each scenario.”

“Phase I: Batch Initialization. The system buffers N frames and performs pose estimation using closed-form solutions: essential matrix decomposition via SVD, EPnP for additional camera poses, and similarity transformation. The complexity is $\mathcal{O}(N)$ for RANSAC-based estimation with a fixed number of cameras. With $N = 2000$ frames in the UAV experiment, this one-time initialization completes in 0.33 ± 0.01 s (mean \pm std. dev. over 10 runs).”

“Phase II: Online Tracking. Each incoming frame requires only fixed-size matrix operations: two-view triangulation via SVD and similarity transformation. The per-frame complexity is $\mathcal{O}(1)$, resulting in an average latency of 0.039 ms per frame. Given the input frequency of 30 FPS (frame interval ≈ 33.3 ms), Phase II achieves a real-time factor of $865\times$, confirming that the system operates well within hard real-time constraints.”

Table 7 Runtime performance comparison between simulation and UAV experiments. All timing statistics are based on 10 repeated experiments.

Metric	Simulation	UAV Experiment
Initialization frames N	300	2000
Phase I initialization time	0.022 ± 0.004 s	0.33 ± 0.01 s
Phase II per-frame latency	0.012 ms	0.039 ms
Real-time factor (30 FPS)	$2882\times$	$865\times$

“As shown in Table 7, both scenarios achieve real-time factors exceeding $800\times$. The timing differences arise from two factors: (1) Phase I processes N accumulated frames, and the UAV experiment uses $N = 2000$ (approximately $6.7\times$ that of simulation’s $N = 300$), resulting in proportionally longer initialization; (2) The UAV experiment operates on real camera imagery requiring detection inference and undistortion, whereas simulation uses synthetic observations with known correspondences.”

Overall Assessment

Comment 6 The paper’s novelty and scientific contribution are limited. The detection component is not operationally viable in real-time settings, and the coordinate recognition pipeline largely reproduces established multi-view geometry methods without comparative validation.

The paper would benefit from: - reformulating the detection method for online inference; - providing detailed comparisons against classical SfM, factorization, and bundle-adjustment baselines; and - clearly positioning the work within existing literature.

Response: We sincerely thank the reviewer for this comprehensive assessment and the constructive suggestions. We have addressed each of these concerns in our revision:

(1) *Online inference capability.* As detailed in our response to Comment 1, we have completely redesigned the system into a dual-phase architecture. Phase I performs offline batch initialization, while Phase II operates as a fully causal, streaming pipeline with per-frame latency of 0.039 ms. This reformulation directly addresses the real-time operational requirement.

(2) *Comparison with classical methods.* As discussed in our response to Comment 2, we have added Supplementary Information 5 providing a systematic analysis of why classical SfM and factorization methods are inapplicable to our problem setting (non-overlapping fields of view, no common feature points). Bundle adjustment is incorporated as an asynchronous refinement module within our framework rather than as a competing baseline.

(3) *Positioning within existing literature.* We have revised the Introduction and Related Work sections to more clearly articulate the unique problem setting addressed by our work: real-time 3D coordinate recognition of non-cooperative airborne targets using position-only ground cameras without overlapping views. Our contribution lies not in proposing new algorithms, but in demonstrating that a carefully designed integration of AI-driven detection and classical geometric techniques can solve this previously unaddressed problem.

We believe these revisions substantially strengthen the paper and address the reviewer’s concerns.

Summary of Changes:

- **Introduction:** Clarified problem setting and contribution positioning.

“To avoid reliance on attitude measurement devices, vision-based methods such as Structure from Motion (SfM) have been developed...In practice, when stationary cameras track a moving target under the background of a featureless sky, conventional SfM becomes inapplicable due to the lack of matchable features.”

- **Algorithm redesign:** Reformulated as dual-phase causal architecture (Phase I: Batch Initialization; Phase II: Online Tracking).

“Phase I: Batch Initialization. The system buffers the initial N frames from all cameras. The YOLOv12-TS module processes the buffered detections using both Outlier Rejection and Trajectory Completion, producing a refined

2D coordinate time series. Phase II: Online Tracking. After initialization, the system performs frame-by-frame processing for real-time coordinate recognition with per-frame latency of 0.039 ms.”

- **Supplementary Section S5:** Added “Applicability Analysis of Conventional Methods” comparing with classical SfM/BA methods.

“Our experimental setup presents a degenerate case for classical methods. SfM fails due to Feature Starvation on Textureless Targets, Rejection of the Dynamic Target, and Stationary Camera Configuration. Pure Global BA is infeasible due to the Initialization Barrier and Latency Barrier ($O(N^3)$ complexity).”

- **Supplementary Section S10:** Added “Computational Complexity Analysis” with timing benchmarks.

“Phase I complexity is $O(N)$, completing in 0.33 ± 0.01 s for $N = 2000$ frames. Phase II complexity is $O(1)$, with per-frame latency of 0.039 ms, achieving real-time factor of $865\times$.”

We sincerely thank the editor and the reviewers once again, and we hope that our responses have sufficiently addressed all of the reviewers’ concerns.

Yours sincerely,

Ke-ke Shang and Michael Small

Computational Communication Collaboratory, Nanjing University, Nanjing, 210093, P.R. China

Complex Systems Group, Department of Mathematics and Statistics, The University of Western Australia, Crawley, Western Australia 6009, Australia

Dear Editors,

We would like to thank you for your prompt processing of our manuscript and the reviewers for furnishing us with very helpful comments and suggestions. We are very pleased with, and encouraged by, their assessment. In the following we respond directly to the comments provided by the referee and describe the changes we have made to the manuscript.

Response to Reviewer 1

The authors addressed all my concerns in a satisfactory manner. I have no further comments.

Response: We sincerely thank you for your very helpful comments and suggestions. We are truly pleased and encouraged by your positive assessment, and we are glad that our revisions have adequately addressed your concerns.

Response to Reviewer 2

Overall, I think the authors have successfully addressed the concerns previously raised. The main remaining issue, requiring a major revision, is now the presentation and organization of the manuscript: introducing a dual-phase / two-stage algorithm is a substantial change, but it currently reads more like an add-on than the result of a thorough rethinking of the method and its exposition.

More concretely, the paper should be reorganized so that the actual contribution is front-and-center. The authors themselves summarize the novelty as: "our novelty is in demonstrating that a carefully designed integration of AI detection and classical techniques can solve this previously unsolved problem setting."

In that spirit, the main text should emphasize the integration - in particular the dual-phase strategy (Algorithm 1) - rather than spending substantial space on detailed derivations of standard formulas.

Response: We sincerely thank the reviewer for this constructive assessment. We fully agree that the dual-phase strategy represents a substantial change to our framework and deserves to be presented as the central contribution rather than an add-on. Following this guidance, we have thoroughly reorganized the manuscript to place the dual-phase strategy front-and-center.

Specifically, in response to the reviewer's comments on presentation and organization, we have made the following key revisions:

1. The former Section 5.5 “Real-time Implementation” has been renamed “Dual-Phase Strategy” and relocated to Section 4.1 as the first subsection of Methods, highlighting its central role in the framework.
2. Algorithm 1, originally in Section 5.5, has been moved to Section 4.1 and significantly expanded: the original version covered only the YOLOv12-TS detection module, whereas the revised algorithm now encompasses the complete dual-phase framework, including multi-camera video input, AI-based detection refinement, and real-time 3D coordinate output.
3. The dual-phase strategy is now introduced prominently in the Introduction, Discussion, and presented as the first subsection of Methods (Section 4.1 “Dual-Phase Strategy”).
4. The numerical simulation in the main text (Section 2.1) now explicitly validates the dual-phase workflow with Phase I/Phase II structure.
5. Detailed mathematical derivations of standard formulas have been moved to Supplementary Methods, keeping only high-level descriptions in the main text. Specifically, we have moved Section 5.3 “Method of coordinate recognition” to the Supplementary Methods, retaining only a high-level description in the main text (now Section 4.2) that explains what the method does, cites key references for standard geometric methods (*Longuet-Higgins 1981*, *Hartley & Zisserman 2003*, *Lepetit et al. 2009*, *Kabsch 1978*, *Prince 2012*), and refers readers to the supplementary material for implementation details.
6. Key ablation studies and computational complexity analysis have been brought into the main text (Section 2.2) to demonstrate the contribution of each component. Specifically, the former Supplementary Section S8.1 “Effect of YOLO-TS Module”, Section S8.2 “Effect of Bundle Adjustment Backend”, and Section S10 “Computational Complexity Analysis” have been moved into Section 2.2 “Coordinate recognition experiment: UAV case” as the “Ablation studies” and “Computational Complexity” subsections.

In addition, we have also reformatted the manuscript according to the editorial guidelines.

We believe these content and structural revisions now properly present our core contribution: the novel integration of AI detection and classical geometric techniques through a dual-phase strategy. All changes have been highlighted in red in the new version.

A few specific suggestions along these lines:

Comment 1 It would make more sense to move section 2 "Numerical simulations of coordinate recognition" to the SM (Supplementary Material), since it pertains to the previous / non-online version of the algorithm. Conversely, move section "S7 Validation of Dual-Phase Strategy via Simulation" from the SM into the main paper, since it validates the revised core strategy.

Response: We thank the reviewer for this insightful suggestion regarding the structural organization of the manuscript. Following this recommendation, we have moved the original Section 2 "Numerical simulations of coordinate recognition", which pertains to the previous non-online version of the algorithm, to the Supplementary Material. Conversely, we have moved Section S7 "Validation of Dual-Phase Strategy via Simulation" from the Supplementary Material into the main text as Section 4.3, since it validates the revised core strategy.

Specifically, the dual-phase real-time simulation (formerly Section S7) is now presented in the main text (Section 2.1), including Phase I/Phase II workflow, performance metrics, and sensitivity analysis (Table 1 and Figure 4). The original offline simulation has been relocated to Supplementary Results, where it serves as further verification that errors under idealized conditions are negligible, confirming the theoretical precision of our SVD-based approach.

Changes:

- **Section 2.1 Numerical simulations:** This section now presents the dual-phase validation (formerly SM Section S7), including Phase I/Phase II workflow, performance metrics, and sensitivity analysis (Table 1 and Figure 4):

"To validate the proposed real-time 3D coordinate recognition method, we conduct numerical simulations under the dual-phase strategy... The trajectory is divided into two phases. During Phase I (Batch Initialization), the first $N = 300$ frames are buffered... After the phase transition, each frame in Phase II (Online Tracking) is processed in real time."

- **Supplementary Results:** Now contains the offline theoretical validation (formerly in main text Section 2) showing sub-millimeter precision under idealized conditions.

- **Introduction:** Added forward reference:

"The theoretical precision of our SVD-based coordinate transformation is further validated through simulation (see Supplementary Results), where errors are negligible under idealized conditions."

Comment 2 Move section "5.3 Method of coordinate recognition" to SM as well.

The equations there appear to be standard multi-view / geometric-method material; this section can be shortened in the main text to a high-level description with appropriate citations, with details deferred to an appendix.

Response: We appreciate this recommendation to streamline the main text. We agree that some of the derivations in this section follow standard multi-view geometry procedures and can be moved to the appendix. Following this suggestion, we have moved Section 5.3 “Method of coordinate recognition” to the Supplementary Methods, retaining only a high-level description in the main text (now Section 4.2) that explains what the method does, cites key references for standard geometric methods (*Longuet-Higgins 1981, Hartley & Zisserman 2003, Lepetit et al. 2009, Kabsch 1978, Prince 2012*), and refers readers to the supplementary material for implementation details.

Specifically, we merged the original three sections, Section 5.1 “Formulation of the coordinate system model”, Section 5.2 “Coordinate transformations”, and Section 5.3 “Method of coordinate recognition” into a single unified Section 4.2 “Coordinate Recognition Framework”. The step-by-step mathematical calculations (coordinate transformations, triangulation derivations, similarity transformation) have been relocated to Supplementary Methods, while the main text now provides only a high-level conceptual description.

Changes:

- **Methods Section 4.2 (Coordinate Recognition Framework):** Merged original Sections 5.1, 5.2, and 5.3 into a single condensed subsection; now contains only high-level description of the approach with explicit reference to Supplementary Methods for detailed derivations. Added key references for standard geometric methods: *Longuet-Higgins 1981, Hartley & Zisserman 2003, Lepetit et al. 2009, Kabsch 1978, Prince 2012*.
- **Supplementary Methods:** Added complete “Method of coordinate recognition” subsection containing all detailed mathematical formulations previously in the main text.

Comment 3 Bring the dual-phase strategy (Algorithm 1 and the key supporting validation/ablation items currently in the SM: S8.1 and S8.2, S10) into the main text, and much earlier than the current placement (certainly not as late as page 21).

Response: We fully agree that the dual-phase strategy should be presented much earlier and more prominently. The former Section 5.5 “Real-time Implementation” has been renamed “Dual-Phase Strategy” and elevated to Section 4.1 as

the first subsection of Methods (page 19), presenting the complete two-phase strategy for coordinate recognition at the very beginning. Algorithm 1 formalizes the entire dual-phase framework and appears prominently at the start of this section. The subsequent subsections—Method of coordinate recognition (Section 4.2), YOLOv12-Time Series (Section 4.3), and Metrics (Section 4.4)—have been streamlined accordingly, with detailed derivations moved to Supplementary Methods.

In addition, the ablation studies and computational complexity analysis (formerly S8.1, S8.2, and S10) have been moved into the main text within Section 2.2 “Coordinate recognition experiment: UAV case”. The new Ablation studies subsection includes Table 2 showing the TS module effectiveness across five YOLO variants (v8n–v12n), and Table 3 comparing trajectory accuracy with and without Bundle Adjustment (RMSE reduced from 6.30 m to 5.45 m for YOLOv12). The new Computational Complexity subsection includes Table 4 with runtime performance, demonstrating Phase I complexity $\mathcal{O}(N)$ and Phase II complexity $\mathcal{O}(1)$.

Changes:

- **Methods Section:** Completely restructured. The former Section 5.5 “Real-time Implementation” has been renamed “Dual-Phase Strategy” and moved to Section 4.1 as the first subsection.
- **Introduction:** Added central contribution statement:

“Specifically, we employ a dual-phase strategy: Phase I (Batch Initialization) accumulates observations to establish precise camera poses, while Phase II (Online Tracking) performs instantaneous coordinate transformation, jointly achieving high accuracy and real-time 3D coordinate recognition.”

- **Results Section 2.2 (UAV experiment):** Added “Ablation studies” subsection containing Table 2 (comprehensive YOLO benchmark with TS module effectiveness across v8–v12), Table 3 (BA backend ablation), and “Computational Complexity” subsection with Table 4 (runtime performance):

“To evaluate the generalizability of the proposed Time Series (TS) module, we tested five YOLO variants: YOLOv8n, YOLOv9t, YOLOv10n, YOLO11n, and YOLOv12n. All models were trained and tested on the same dataset (Cameras 1-3) with identical hyperparameters.”

“Phase I: Batch Initialization. The system buffers N frames and performs pose estimation using closed-form solutions... The complexity is $\mathcal{O}(N)$ for

RANSAC-based estimation.”

“Phase II: Online Tracking. Each incoming frame requires only fixed-size matrix operations: two-view triangulation via SVD and similarity transformation. The per-frame complexity is $\mathcal{O}(1)$, resulting in an average latency of 0.039 ms per frame.”

- **Algorithm 1:** Now presented in Methods Section 4.1 (Dual-Phase Strategy), describing the complete dual-phase framework for real-time 3D coordinate recognition. The algorithm integrates both AI detection refinement (YOLOv12-TS) and geometric transformation, covering the full workflow from video input to 3D world coordinates output.

Comment 4 Section ”5.6 Metrics” feels superfluous: the metrics listed are standard, so this section could be significantly shortened or moved to SM (or reduced to a brief statement with citations).

Response: We agree with this suggestion. Since readers familiar with the field will recognize these metrics, a brief statement in the main text is sufficient. We have shortened the Metrics section (now Section 4.4) to concisely list the metrics used, while moving the detailed mathematical definitions to Supplementary Methods for reference.

Changes:

- **Methods Section 4.4 (Metrics):** Shortened from detailed equations to two brief paragraphs:

“Metrics for 2D object detection. We evaluate the performance of 2D detection using Precision (IoU-P), Recall (IoU-R), and F1-score (IoU-F1) computed at different IoU thresholds... These metrics assess the accuracy of predicted bounding boxes by measuring the overlap with ground-truth annotations (Supplementary Methods).”

“Metrics for 3D coordinate recognition. We evaluate the performance of 3D coordinate recognition using four metrics: Root Mean Square Error (RMSE), Mean Absolute Error (MAE), Maximum Error, and R-squared (R^2)... Detailed definitions are provided in Supplementary Methods.”

- **Supplementary Methods:** Added “Metrics Details” subsection containing complete mathematical definitions of RMSE, MAE, Maximum Error, R^2 , IoU-P, IoU-R, and IoU-F1 with all formulas.

Comment 5 More generally, as it stands, too much of the manuscript’s scientific

“payload” appears to sit in the SM rather than in the main text, which is unusual and weakens the presentation.

Response: We thank the reviewer for this important observation regarding the distribution of scientific content. We fully agree that the previous version placed too much core content in the Supplementary Material. To address this, we have restructured the manuscript to give central prominence to our dual-phase strategy: the former Section 5.5 “Real-time Implementation” has been renamed and relocated to Section 4.1 “Dual-Phase Strategy” as the first subsection under Methods.

Accordingly, we have expanded Algorithm 1 to formalize the complete two-phase framework, which integrate AI detection, batch initialization, and online tracking, making it the central reference point throughout the manuscript. Algorithm 1 is now cited in the Introduction, Section 2.1 (Numerical simulations), Section 4.1 (Dual-Phase Strategy), Section 4.2 (Coordinate Recognition Framework), and Section 4.3 (YOLOv12-TS).

Following the reviewer’s guidance, we have rebalanced the distribution of content between the main text and Supplementary Material according to scientific importance. The dual-phase strategy, which enables real-time performance, is now presented prominently in the manuscript.

We sincerely thank the editor and the reviewers once again, and we hope that our responses have sufficiently addressed all concerns.

Yours sincerely,

Ke-ke Shang and Michael Small

Computational Communication Collaboratory, Nanjing University, Nanjing, 210093, P.R. China

Complex Systems Group, Department of Mathematics and Statistics, The University of Western Australia, Crawley, Western Australia 6009, Australia